# Pareto-Optimal Energy Alignment for Designing Nature-Like Antibodies

**Yibo Wen**   **Chenwei Xu**   **Jerry Yao-Chieh Hu**   **Kaize Ding**   **Han Liu**

Northwestern University

{`yibo`,`cwx`,`jhu`}@u.northwestern.edu, {`kaize.ding`,`hanliu`}@northwestern.edu

## Abstract

We present a three-stage framework for training deep learning models specializing in antibody sequence-structure co-design. We first pre-train a language model using millions of antibody sequence data. Then, we employ the learned representations to guide the training of a diffusion model for joint optimization over both sequence and structure of antibodies. During the final alignment stage, we optimize the model to favor antibodies with low repulsion and high attraction to the antigen binding site, enhancing the rationality and functionality of the designs. To mitigate conflicting energy preferences, we extend AbDPO (Antibody Direct Preference Optimization) to guide the model toward Pareto optimality under multiple energy-based alignment objectives. Furthermore, we adopt an iterative learning paradigm with temperature scaling, enabling the model to benefit from diverse online datasets without requiring additional data. In practice, our proposed methods achieve high stability and efficiency in producing a better Pareto front of antibody designs compared to top samples generated by baselines and previous alignment techniques. Through extensive experiments, we showcase the superior performance of our methods in generating nature-like antibodies with high binding affinity.

## 1   Introduction

Antibodies are large, Y-shaped proteins that play a crucial role in protecting the human body against various disease-causing antigens [Scott et al., 2012]. As shown in Figure 1, an antibody consists of two identical heavy chains and two identical light chains. Antibodies possess remarkable abilities to bind a wide range of antigens, and the tips of the Y shape exhibit the most variability [Collis et al., 2003, Chiu et al., 2019]. These critical regions, composed of specific arrangements of amino acids, are known as Complementarity Determining Regions (CDRs) since their shapes complement those of antigens. To a great extent, the CDRs at the tips of light and heavy chains determine an antibody's specificity to antigens [Akbar et al., 2021]. Hence, the key challenge in antibody design is identifying and designing effective CDRs as part of the antibody framework that bind to specific antigens.

Recently, various deep learning models achieve great success in the long-standing problem of antibody design and optimization. For example, Madani et al. [2023] and Rives et al. [2019] borrow ideas from language models and treat proteins as sequences to predict their structures, functions, and other important properties. These methods benefit from having access to large datasets with millions of protein sequences, but often lead to subpar results in generation tasks conditioned on protein structures [Gao et al., 2023, Martinkus et al., 2024]. Due to the determinant role of structure in protein function, co-designing sequences with structures emerges as a more promising approach [Anishchenko et al., 2020, Harteveld et al., 2022, Jin et al., 2022a,b]. Among all, diffusion-based

---

Future updates can be found at `https://arxiv.org/abs/2412.20984`.

39th Conference on Neural Information Processing Systems (NeurIPS 2025).

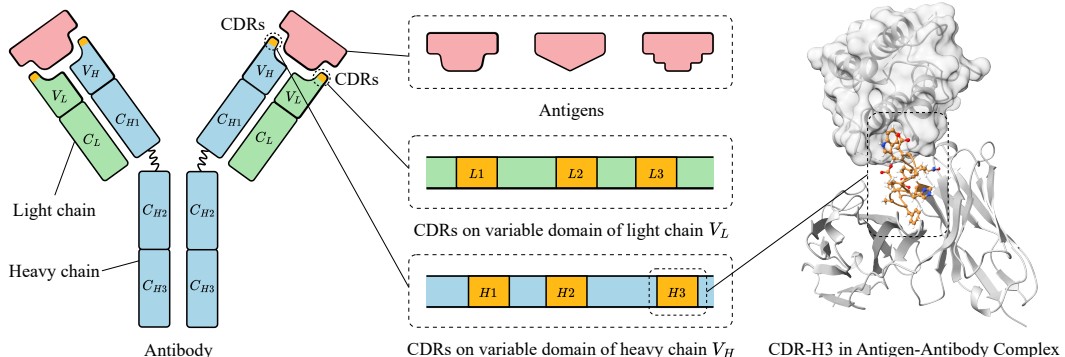

Figure 1: Illustration of an antibody binding to an antigen. The antibody's light and heavy chains are shown with their variable (V) and constant (C) regions. The third CDR in the heavy chain (CDR-H3), colored in orange, is critical for determining the binding affinity to the antigen.

methods stand out by learning the reverse process of transforming desired protein structures from noise [Vinod, 2022, Lisanza et al., 2023, Martinkus et al., 2024]. These methods achieve atomic-resolution antibody design and state-of-the-art results in various tasks, including sequence-structure co-design, fix-backbone CDR design, and antibody optimization [Luo et al., 2022, Zhou et al., 2024b].

Despite the prevalence of generative models, two key problems persist in effective antibody sequence-structure co-design. **First**, datasets containing complete 3D structures of antibodies are orders of magnitude smaller than sequence-only datasets. For example, the most common dataset for antibody design, SAbDab [Dunbar et al., 2014], only contains a few thousand antibody structures despite daily updates. The scarcity of high-quality antigen-antibody pairs, coupled with high variability of CDR structures [Collis et al., 2003], further constrains the performance of learning-based approaches. **Second**, existing methods overlook energy functions during supervised training and struggle to generate antibodies with low repulsion and high binding affinity. Contrary to traditional computational methods, recent efforts [Luo et al., 2022, Jin et al., 2022a,b, Kong et al., 2022] shift their focus from searching for minimal energy states to optimizing metrics such as Amino Acid Recovery (AAR) and Root Mean Square Deviation (RMSD). However, these metrics are prone to manipulation, often fail to differentiate between different error types, and ignore important side chain structures in CDR-antigen interactions [Zhou et al., 2024b]. Overreliance on these metrics gives rise to irrationality in generated structures and widens the gap between *in silico* and *in vitro* antibody design.

To address these challenges, we introduce a three-stage training pipeline that focuses on the rationality and functionality of the antibody. Inspired by the recent success of Large Language Models, we adopt a similar training paradigm comprising pre-training, transferring and alignment.

1. **Pre-training.** We first utilize a pre-trained antibody language model, trained on millions of amino acid sequences, to alleviate the shortage of structured antibody data. This approach enables the model to capture underlying relationships between proteins and internalize fundamental biological concepts such as structure and function [Rives et al., 2019, Chowdhury et al., 2021].

2. **Transferring.** We use the language model's learned representations to train a smaller model on a curated antibody-antigen dataset, adapting it for antigen-specific antibody design. The diffusion-based model is able to recover not only sequences but also coordinates and side-chain orientations of each amino acid conditioned on the antigen-antibody framework [Luo et al., 2022].

3. **Alignment.** For the final stage, we conduct energy-based alignment of the diffusion model using Pareto-Optimal Energy Alignment as an extension of Direct Preference Optimization (DPO) [Wallace et al., 2023]. By reusing designs generated by the model and labeling them with biophysical energy measurements, we compel the model to favor antibodies with lower repulsion and higher affinity in a data-free fashion. Additionally, we introduce an iterative version of the alignment algorithm in an online setting, allowing the model to benefit from online exploration. To balance exploration and exploitation during alignment, we propose decaying temperature scaling during the sampling process. Empirical results verify that our methods surpass existing alignment methods, consistently generating antibodies with energies closer to Pareto optimality.

In summary, our main contributions are as follows:

- We devise the first three-stage training framework for antibody sequence-structure co-design, consisting of pre-training, transferring, and alignment.

- We propose an efficient multi-objective alignment algorithm with online exploration which consistently produces a better Pareto front of models in terms of energy without extra data.

- Our approach achieves state-of-the-art performance in generating more natural-like antibodies with better rationality and functionality.

## 2 Related Work

**Computational Antibody Design.** Deep learning methods are widely used for antibody design, with many latest works incorporating generative models [Alley et al., 2019, Saka et al., 2021, Shin et al., 2021, Akbar et al., 2022]. Jin et al. [2022a] introduce HERN, a hierarchical message-passing network for encoding atoms and residues. Kong et al. [2022] propose MEAN, utilizing E(3)-equivariant graph networks to better capture the geometrical correlation between different components. Additionally, Kong et al. [2023] propose dyMEAN, focusing on epitope-binding CDR-H3 design and modeling full-atom geometry. Luo et al. [2022] propose a diffusion model that uses residue type, atom coordinates, and side-chain orientations to generate antigen-specific CDRs. Martinkus et al. [2024] propose Ab-Diffuser, which incorporates more domain knowledge and physics-based constraints.

**Diffusion-based Generative Models.** Diffusion models are a type of generative model with an encoder-decoder structure. It involves a Markov-chain process with diffusion steps to add noise to data (encoder) and reverse steps to reconstruct desired data from noise (decoder) [Weng, 2021, Luo, 2022, Chan, 2024]. DDPM [Ho et al., 2020] is one of the most well-known diffusion models utilizing this process. Song et al. [2020a] propose DDIM, which is an improved version of DDPM that reduces the number of steps in the generation process. Score-matching [Hyvärinen and Dayan, 2005, Vincent, 2011, Song et al., 2020b] is also a popular research area in diffusion models. The key idea of score-matching is to use Langevin dynamics to generate samples and estimate the gradient of the data distribution. Later, Song et al. [2021] propose a solver for faster sampling in the context of score-matching methods using stochastic differential equations.

**Alignment of Generative Models.** Preference alignment during fine-tuning improves the quality and usability of generated data. Reinforcement Learning (RL) is one popular approach to align models with human preferences, and RLHF [Ouyang et al., 2022] is an example of such algorithm. Rafailov et al. [2024] propose DPO as an alternative approach to align with human preferences. Different from RL-based approaches, DPO achieves higher stability and efficiency as it does not require explicit reward modeling. Building upon DPO, recent works such as DDPO [Black et al., 2023], DPOK [Fan et al., 2024], and DiffAC [Zhou et al., 2024a] demonstrate the possibility of adapting existing alignment techniques to various generative models. SimPO [Meng et al., 2024] improves DPO by using the average log probability of a sequence as the implicit reward.

## 3 Preliminaries

**Problem Definition.** In this work, we represent each amino acid by its residue type $s_i \in$ {ACDEFGHIKLMNPQRSTVWY}, coordinate $\boldsymbol{x}_i \in \mathbb{R}^3$, and orientation $\boldsymbol{O}_i \in \mathrm{SO}(3)$, where $i \in \{1, \ldots, N\}$. Here, $N$ is the total number of amino acids in the protein complex, which may contain multiple chains [Luo et al., 2022]. We focus on designing CDR, a critical functioning component of the antibody, given the remaining antibody and antigen structure. Let the CDR of interest consists of $m$ amino acids starting from index $l+1$ to $l+m$ on the entire antibody-antigen framework with a total of $N$ amino acids. We denote the target CDR as $\mathcal{R} = \{(s_j, \boldsymbol{x}_j, \boldsymbol{O}_j) \mid j = l+1, \ldots, l+m\}$ and the given antibody-antigen framework as $\mathcal{F} = \{(s_i, \boldsymbol{x}_i, \boldsymbol{O}_i) \mid i \in \{1, \ldots, N\}\setminus\{l+1, \ldots, l+m\}\}$. Therefore, our objective is to model the conditional distribution $P(\mathcal{R} \mid \mathcal{F})$.

**Direct Preference Optimization.** To tackle the common issues of fine-tuning with Reinforcement Learning (RL), Rafailov et al. [2024] propose DPO as an alternative for effective model alignment. In the setting of DPO, we have an input $x$ and a pair of outputs $(y_1, y_2)$ from dataset $\mathcal{D}$, and a corresponding preference denoted as $y_w \succ y_l \mid x$ where $y_w$ and $y_l$ are the "winning" and "losing" samples amongst $(y_1, y_2)$ respectively. According to the Bradley-Terry (BT) model [Bradley and Terry, 1952], for a pair of outputs, the human preferences are governed by a ground truth reward

model $r(x, y)$ such that BT preference model is

$$p(y_1 \succ y_2 \mid x) = \sigma(r(x, y_1) - r(x, y_2)), \qquad (3.1)$$

where $\sigma(\cdot)$ is sigmoid. Then, the optimal policy $\pi_r^*$ is defined by maximizing reward:

$$\pi_r^* = \underset{\pi}{\arg\max} \; \underset{x \sim \mathcal{D}, y \sim \pi(y|x)}{\mathbb{E}} \left[ r(x, y) - \beta \log \frac{\pi(y \mid x)}{\pi_{\text{ref}}(y \mid x)} \right], \qquad (3.2)$$

where $\beta$ is the inverse temperature controlling the KL regularization. By solving (3.2) analytically, Rafailov et al. [2024] give a relation between the ground-truth reward and optimal policy:

$$r(x, y) = \beta \log \frac{\pi_r^*(y \mid x)}{\pi_{\text{ref}}(y \mid x)} + \beta \log Z(x), \qquad (3.3)$$

where $Z(x) = \sum_y \pi_{\text{ref}}(y \mid x) \exp\left(r(x, y)/\beta\right)$. This allows us to rewrite BT preference model (3.1) without reward model $r$ (only in $\pi_r^*, \pi_{\text{ref}}$):

$$p(y_w \succ y_l \mid x) = \sigma\left( \beta \log \frac{\pi_r^*(y_w \mid x)}{\pi_{\text{ref}}(y_w \mid x)} - \beta \log \frac{\pi_r^*(y_l \mid x)}{\pi_{\text{ref}}(y_l \mid x)} \right). \qquad (3.4)$$

In this way, the maximum likelihood reward objective for a parameterized policy $\pi_\theta$ becomes:

$$\mathcal{L}_{\text{DPO}}(\pi_\theta; \pi_{\text{ref}}) = - \underset{(x, y_w, y_l) \sim \mathcal{D}}{\mathbb{E}} \left[ \log \sigma \left( \beta \log \frac{\pi_\theta(y_w \mid x)}{\pi_{\text{ref}}(y_w \mid x)} - \beta \log \frac{\pi_\theta(y_l \mid x)}{\pi_{\text{ref}}(y_l \mid x)} \right) \right]. \qquad (3.5)$$

This derived loss function bypasses the need for explicit reward modeling, enabling an RL-free approach for preference optimization. While DPO is first designed for language models, we are able to re-formulate it for diffusion models and arrive at a similar differentiable objective following [Wallace et al., 2023], or see Appendix A.3 for details.

## 4 Methodology

In this section, we present our energy alignment method for designing nature-like antibodies, named **AlignAb**. We introduce Pareto-Optimal Energy Alignment to fine-tune the model under conflicting energy preferences in Section 4.1. Then, we present an iterative version of the algorithm and discuss how to mitigate mode collapse during sampling with temperature scaling in Section 4.2. Finally, we summarize the alignment algorithm and three-stage training framework in Section 4.3.

### 4.1 Pareto-Optimal Energy Alignment (POEA)

Pre-trained models often struggle to produce natural-like antibodies because they tend to ignore important physical properties during the optimization process. These physical properties manifest themselves as various energy measurements such as Lennard-Jones potentials (accounting for attractive and repulsive forces), Coulombic electrostatic potential, and hydrogen bonding energies [Adolf-Bryfogle et al., 2017]. We aim to close this gap by aligning the pre-trained model to favor antibodies with low repulsion and high attraction energy configurations at the binding site. While AbDPO [Zhou et al., 2024b] demonstrates the potential of naïve DPO in antibody design, there are two primary distinctions in this context:

(D1) The ground-truth reward model, given by energy measurements, is available.

(D2) There are multiple, often conflicting, energy-based preferences.

Therefore, we propose Pareto-Optimal Energy Alignment to address (D1) by adding ground-truth reward margin, and (D2) by extending DPO to multiple preferences.

**Incorporating Reward Model.** Since we have access to the ground-truth reward model, it would be unwise to ignore this extra information and perform alignment with just the preference labels. We show how to extend DPO and incorporate the available reward values in the training objective. Let's consider a new reward function $r'(x, y) := r(x, y) + f(x)$ by adding the ground-truth reward model $r(x, y)$ and a random reward model $f(x)$ which depends only on the input. According to (3.3), we express $r'(x, y)$ in terms of its optimal policy under the KL constraint:

$$r'(x, y) = \beta \log \frac{\pi_{r'}^*(y \mid x)}{\pi_{\text{ref}}(y \mid x)} + \beta \log Z(x), \qquad (4.1)$$

where $Z(x) = \sum_y \pi_{\text{ref}}(y \mid x) \exp\left(r'(x,y)/\beta\right)$. Note that $r'(x,y)$ and $r(x,y)$ induce the same optimal policy by construction (see Lemma A.2 and Appendix A.2 for details):

$$\pi^*_{r'} = \pi^*_r = \underset{\pi}{\text{argmax}} \; \underset{x \sim \mathcal{D}, y \sim \pi(y|x)}{\mathbb{E}} \left[ r(x,y) - \beta \log \frac{\pi(y \mid x)}{\pi_{\text{ref}}(y \mid x)} \right].$$

Then, we cast the random reward model $f(x)$ into a function of $\pi^*_r$ and $r$:

$$f(x) = \beta \log \frac{\pi^*_r(y \mid x)}{\pi_{\text{ref}}(y \mid x)} + \beta \log Z(x) - r(x,y). \tag{4.2}$$

Finally, we replace $r(x,y)$ with $f(x)$ in the original preference model $p(y_1 \succ y_2 \mid x) = \sigma(r(x,y_1) - r(x,y_2))$ and hence DPO loss (3.5) becomes below loss over the parametrized model $\pi_\theta$) as

$$-\underset{(x,y_w,y_l) \sim \mathcal{D}}{\mathbb{E}} \left[ \log \sigma \left( \beta \log \frac{\pi_\theta(y_w \mid x)}{\pi_{\text{ref}}(y_w \mid x)} - \beta \log \frac{\pi_\theta(y_l \mid x)}{\pi_{\text{ref}}(y_l \mid x)} - \Delta_r \right) \right], \tag{4.3}$$

where $\Delta_r := r(x,y_w) - r(x,y_l)$ is the positive reward margin between $y_w$ and $y_l$. Notably, the obtained loss differs from the vanilla DPO loss (3.5) by including an additional reward margin $\Delta_r$. To better understand how the derived loss facilitates the alignment process, we take the gradient of the loss and interpret each term individually:

$$-\beta \underset{(x,y_w,y_l) \sim \mathcal{D}}{\mathbb{E}} \left[ \sigma\big(\underbrace{\tilde{r}_\theta(x,y_l) - \tilde{r}_\theta(x,y_w) + \Delta_r}_{\text{(I): combined sample weight}}\big) \cdot \Big[ \underbrace{\nabla_\theta \log \pi(y_w \mid x)}_{\text{(II): increase likelihood of } y_w} - \underbrace{\nabla_\theta \log \pi(y_l \mid x)}_{\text{(III): decrease likelihood of } y_l} \Big] \right],$$

where $\tilde{r}_\theta(x,y) = \beta \log \frac{\pi_\theta(y|x)}{\pi_{\text{ref}}(y|x)}$ is the implicit reward defined by the models. Similar to the DPO gradient, term (II) and (III) aim to increase the likelihood of the preferred sample $y_w$ and decrease that of the dispreferred sample $y_l$. The key difference lies in term (I), where our weighting incorporates both the implicit reward margin $\tilde{r}\theta(x,y_w) - \tilde{r}\theta(x,y_l)$ and the explicit ground-truth reward margin $\Delta_r$, ensuring larger reward gaps lead to stronger weight adjustments.

**Multi-Objective Alignment.** Given $n$ ground-truth reward models $\boldsymbol{r} = [r_1, \ldots, r_n]^\mathsf{T}$, we construct a dataset $\hat{\mathcal{D}} = \{(x_i, y_i, \boldsymbol{r}(x,y_i))\}$ that records the reward values for each input and its corresponding output. In practice, each reward value is an energy measurement associated with certain physical properties. Following Zhou et al. [2023], the goal for multi-objective preference alignment is not to learn a single optimal model but rather a Pareto front of models $\{\pi^*_{\hat{r}} \mid \hat{r} = \boldsymbol{w}^\mathsf{T} \boldsymbol{r}, \boldsymbol{w} \in \Omega\}$ and each solution optimizes for one specific collective reward model $\hat{r}$:

$$\pi^*_{\hat{r}} = \underset{\pi}{\text{argmax}} \; \underset{x,y \sim \hat{\mathcal{D}}}{\mathbb{E}} \left[ \hat{r}(x,y) - \beta \log \frac{\pi(y \mid x)}{\pi_{\text{ref}}(y \mid x)} \right], \tag{4.4}$$

where $\boldsymbol{w} = [w_1, \ldots, w_n]^\mathsf{T}$ s.t. $\sum_{i=1}^n w_i = 1$ is a weighting vector in the preference space $\Omega$. To obtain a preference pair $(x, y_w, y_l)$, we first select two random data points $(x, y_i, \boldsymbol{r}(x,y_i))$ and $(x, y_j, \boldsymbol{r}(x,y_j))$ from $\hat{\mathcal{D}}$ and then compute their collective rewards $\hat{r}(x,y_i)$ and $\hat{r}(x,y_j)$. Among $(y_i, y_j)$, we assign $y_w \succ y_l \mid x$ which satisfies $\hat{r}(x,y_w) > \hat{r}(x,y_l)$.

To incorporate multiple preferences, we replace the original reward model $r$ in (4.3) with the collective reward model $\hat{r} = \boldsymbol{w}^\mathsf{T} \boldsymbol{r}$ and arrive at a **P**areto-**O**ptimal-**E**nergy-**A**lignment (**POEA**) loss:

$$\mathcal{L}_{\text{POEA}}(\pi_\theta; \pi_{\text{ref}}) = -\underset{(x,y_w,y_l) \sim \hat{\mathcal{D}}}{\mathbb{E}} \left[ \log \sigma \big( \beta \log \frac{\pi_\theta(y_w \mid x)}{\pi_{\text{ref}}(y_w \mid x)} - \beta \log \frac{\pi_\theta(y_l \mid x)}{\pi_{\text{ref}}(y_l \mid x)} - \Delta_{\hat{r}} \big) \right], \tag{4.5}$$

where $\Delta_{\hat{r}} := \hat{r}(x,y_w) - \hat{r}(x,y_w)$. This simple formulation inherits the desired properties from its single-objective counterpart, ensuring that it produces the optimal model $\pi_{\hat{r}}$ for each specific $\boldsymbol{w}$. In practice, we calculate the reward margin with energy measurements following Equation (D.4).

### 4.2 Iterative Alignment with Temperature Scaling

**Iterative Online Alignment.** To further exploit the available reward model, we develop an iterative version of our alignment method as an analogy to online reinforcement learning (RL). Instead of relying on a large offline dataset collected prior to training as in AbDPO [Zhou et al., 2024b], our approach starts with an empty dataset and augments it with an online dataset constructed by querying the current model at the start of each iteration. This method mirrors how online RL agents gather

data and learn by interacting with the environment, enabling continuous policy improvement. We present the detailed algorithm in Algorithm 1. Ideally, we are able to repeat the process until no further improvement is observed, and we select the best model based on validation metrics.

**Temperature Scaling.** While CDRs exhibit substantial sequence variation across antibodies [Collis et al., 2003], parameterized neural networks often struggle to capture this diversity and suffer from mode collapse during training [Bayat, 2023]. To quantify this effect, we measure the entropy of generated sequences, defined as $H = -\sum p \log p$, and compare it with that of natural CDR-H3 sequences. As shown in Table 1, a clear entropy gap emerges, indicating reduced diversity and potential model collapse, particularly evident when comparing results at different training steps.

To mitigate this issue, we introduce temperature scaling during inference. This technique adjusts the logits prior to the softmax operation to modulate sampling randomness (i.e., entropy). Specifically, the scaled softmax is given by $\text{Softmax}(z_i/T) = \frac{\exp(z_i/T)}{\sum_j \exp(z_j/T)}$, where $T$ denotes the temperature. A higher $T$ increases diversity, while a lower $T$ promotes determinism. Since our diffusion model uses multinomial distribution to model antibody sequences (see Appendix A.1), we inject a small temperature factor at inference time to enhance sequence diversity. Inspired by

Table 1: CDR-H3 Entropy

| Method | Entropy ($\uparrow$) |
|---|---|
| Reference | 3.95 |
| MEAN | 2.18 |
| DiffAb (100k step) | 3.57 |
| DiffAb (200k step) | 3.29 |
| **DiffAb-TS** | **3.84** |

the $\varepsilon$-greedy strategy in RL, we adopt a decaying temperature schedule to balance exploration and exploitation. We validate this approach by applying a small temperature scale ($T = 1.5$) to the pre-trained diffusion model DiffAb [Luo et al., 2022]. The resulting model, DiffAb-TS, produces sequences that match the diversity of natural CDR-H3 sequences, as shown in Table 1.

---

**Algorithm 1** Iterative Pareto-Optimal Energy Alignment

---

1: **Input:** Initial dataset $\hat{\mathcal{D}}_0 = \emptyset$, KL regularization coefficient $\beta$, total online iterations $T$, batch size $m$, reference model $\pi_{\text{ref}}$, initial model $\pi_0 = \pi_{\text{ref}}$, and reward model $\hat{r}$.
2: **for** $t = 0, 1, 2, \cdots, T$ **do**
3:     Sample input prompts $x_i \sim \mathcal{X}$ for $i = 1, \ldots, m$.
4:     Generate two responses for each prompt: $y_i^{(1)}, y_i^{(2)} \sim \pi_t(\cdot \mid x_i)$.
5:     Calculate rewards $\hat{r}(x_i, y_i^{(1)})$ and $\hat{r}(x_i, y_i^{(2)})$ for all $i \in [m]$, and collect them as $\hat{\mathcal{D}}_t$.
6:     Optimize $\pi_{t+1}$ with $\hat{\mathcal{D}}_{0:t}$ according to (4.5):

$$\pi_{t+1} \leftarrow \operatorname*{argmin}_{\pi} \mathbb{E}_{(x, y_w, y_l) \sim \hat{D}_{0:t}} \left[ \log \sigma \left( \beta \log \frac{\pi_\theta(y_w \mid x)}{\pi_{\text{ref}}(y_w \mid x)} - \beta \log \frac{\pi_\theta(y_l \mid x)}{\pi_{\text{ref}}(y_l \mid x)} - \Delta_{\hat{r}} \right) \right].$$

7: **end for**
8: **Output:** Best-performing policy $\pi_{t^*}$ selected from $\{\pi_0, \pi_1, \ldots, \pi_T\}$ using a validation set.

---

### 4.3 Three-Stage Training Framework

Inspired by the recent success of large language models, we adapt the widely used three-stage training framework to the task of antibody design in combination with our devised alignment method.

- **Pre-training.** Due to the limited availability of structured antibody data, we leverage the abundant online antibody sequences for pre-training using a BERT-based model [Devlin et al., 2019]. Following Gao et al. [2023], we employ a masked language modeling objective, where we mask all residues within CDRs and aim to recover them. This approach enables the antibody language model to learn expressive representations that capture the underlying relationships between proteins and internalize fundamental biological concepts such as structure and function.

- **Transferring.** We use the pretrained BERT model as a frozen encoder to train a downstream diffusion model. Specifically, this transfers learned representations to the diffusion model for antibody generation (see details of embedding fusion in Appendix E.1). Crucially, this representation enhancement addresses the challenge of antigen-specific antibody design: datasets are limited and curated by human experts. The diffusion-based model recovers sequences, coordinates, and orientations of each amino acid, conditioned on the entire antigen-antibody framework. For detailed formulation on diffusion models for antibody generation, see Appendix A.1.

- **Alignment.** Lastly, we align the trained diffusion model via Pareto-Optimal-Energy-Alignment (POEA) from (3.2), an extended version of multi-objective DPO-diffusion for antibody design. Importantly, the Pareto weight **w** allows us to incorporate designers' preferences, enabling balanced control over multiple objectives (physical, chemical, and biological properties) by domain experts. In summary, we propose POEA (3.2) to address issues of conflicting energy preferences and potential mode collapse during the alignment stage. We take advantage of ground-truth reward models (see detailed reward calculations in Appendix D) by incorporating reward margin in the loss function and utilizing online exploration datasets.

## 5 Experimental Studies

We evaluate our proposed framework, **AlignAb**, on the task of designing antigen-binding CDR-H3 regions. In Section 5.1, we outline the experimental setup for the three training stages. We then introduce the evaluation metrics and discuss the main results in Section 5.2, followed by comprehensive ablation studies in Section 5.3.

### 5.1 Experiment Setup

**Energy Definitions.** We introduce four key energy measurements where we use the first two to evaluate the rationality and functionality of antibodies and use the rest to generate preferences during alignment. To determine the rationality and functionality of different CDR designs, we identify two key energy measurements: CDR $E_{\text{total}}$ and CDR-Ag $\Delta G$.

(1) CDR $E_{\text{total}}$ represents the combined energy of all amino acids within CDR, calculated using the default score function in Rosetta [Chaudhury et al., 2010]. This energy is a strong indicator of structural rationality, as higher $E_{\text{total}}$ suggests large clashes between residues.

(2) CDR-Ag $\Delta G$ represents the binding energy between the CDR and the antigen, determined using the protein interface analyzer in Rosetta [Chaudhury et al., 2010]. This measurement reflects the difference in total energy when the antibody is separated from the antigen. Lower $\Delta G$ corresponds to higher binding affinity, serving as a strong indicator of structural functionality.

To generate energy-based preferences during model alignment, we use two fine-grained energy measurements: CDR-Ag $E_{\text{rep}}$ and CDR-Ag $E_{\text{att}}$.

(3) CDR-Ag $E_{\text{att}}$ captures the attraction forces between the designed CDR and the antigen.

(4) CDR-Ag $E_{\text{rep}}$ captures the repulsion forces between the designed CDR and the antigen.

As suggested by Zhou et al. [2024b], we further decompose $E_{\text{att}}$ and $E_{\text{rep}}$ at the amino acid level to provide more explicit and intuitive gradients. We include detailed calculation formulas for the energy measurements and their corresponding reward functions in Appendix D. We exclude CDR $E_{\text{total}}$ and CDR-Ag $\Delta G$ measurements when determining the preference pairs because our experiments demonstrate that CDR-Ag $E_{\text{att}}$ and CDR-Ag $E_{\text{rep}}$ are sufficient for effective model alignment. This simplification reduces the computational cost associated with tuning multiple weights for different reward models, resulting in a more efficient and stable alignment process.

**Datasets.** For pre-training, we utilize the antibody sequence data from the Observed Antibody Space database [Olsen et al., 2022]. Following Gao et al. [2023], we adopt the same preprocessing steps including sequence filtering and clustering. Since we focus on CDR-H3 design, we select 50 million heavy chain sequences to pre-train the model.

To transfer the knowledge, we use the antibody-antigen data with structural information from SAbDab database [Dunbar et al., 2014]. Following Kong et al. [2022], we first remove complexes with a resolution worse than 4Å and renumber the sequences under the Chothia scheme [Chothia and Lesk, 1987]. Then, we identify and collect structures with valid heavy chains and protein antigens. We also discard duplicate data with the same CDR-H3 and CDR-L3. We use MMseqs2 [Steinegger and Söding, 2017] to cluster the remaining complexes with a threshold of 40% sequence similarity based on the CDR-H3 sequence of each complex. During training, we split the clusters into a training set of 2,340 clusters and a validation set of 233 clusters. For testing, we borrow the RAbD benchmark [Adolf-Bryfogle et al., 2017] and select 42 legal complexes not used in training.

For alignment, we avoid using additional datasets and only draw samples from the trained diffusion model. During each iteration, we first generate 1,280 unique CDR-H3 designs and collect them as the online dataset. Then, we reconstruct the full CDR structure including side chains at the atomic level

Table 2: Summary of CDR $E_{\text{total}}$, CDR-Ag $\Delta G$, CDR-Ag $E_{\text{att}}$, and CDR $E_{\text{rep}}$ (kcal/mol) of reference antibodies, ranked top-1 antibodies and total antibodies designed by our model and other baselines (MEAN, HERN, dyMEAN, ABGNN, DiffAb, AbX). We compute the generation gap as the mean absolute error relative to the reference. Lower values are better in all measurements. Our results show that our generated antibodies are closer to references compared to all baseline methods.

| Method | CDR $E_{\text{total}}$ | | CDR-Ag $\Delta G$ | | CDR-Ag $E_{\text{att}}$ | | CDR-Ag $E_{\text{rep}}$ | | Gap | |
|---|---|---|---|---|---|---|---|---|---|---|
| | Top | Avg. | Top | Avg. | Top | Avg. | Top | Avg. | Top | Avg. |
| Reference | -19.33 | - | -16.00 | - | -18.34 | - | 18.05 | - | - | - |
| MEAN | 46.27 | 186.05 | **-19.94** | 26.14 | -5.13 | -5.16 | 7.77 | 29.21 | 31.16 | 73.14 |
| HERN | 7,345.11 | 10,599.92 | 640.50 | 2,795.15 | -6.64 | -1.98 | **1.67** | 36.88 | 1453.75 | 2416.97 |
| dyMEAN | 5,074.11 | 12,311.15 | 4,452.26 | 10,881.22 | -12.62 | -5.06 | 139.42 | 1,762.59 | 2422.10 | 6183.425 |
| ABGNN | 1315.34 | 3022.88 | -11.52 | **16.08** | -1.63 | -0.48 | 22.15 | **8.84** | 354.38 | 778.54 |
| DiffAb | -1.50 | 158.90 | -6.18 | 260.30 | -12.30 | **-15.71** | 18.63 | 603.58 | 19.74 | 263.44 |
| AbX | 13.56 | **25.33** | 94.76 | 170.34 | -13.68 | -15.12 | 21.38 | 38.20 | 37.75 | 63.43 |
| **AlignAb** | **-6.37** | 30.45 | -8.81 | 25.16 | **-14.89** | -14.81 | 15.52 | 56.22 | **17.91** | **39.00** |

using PyRosetta [Chaudhury et al., 2010], and record the predefined energies for each CDR at residue level. We repeat this iterative process 3 times for each antibody-antigen complex in the test set.

**Baselines.** We compare AlignAb with 5 recent state-of-the-art antibody sequence-structure co-design baselines. **MEAN** [Kong et al., 2022] generates sequences and structures using a progressive full-shot approach. **HERN** [Jin et al., 2022a] generates sequences autoregressively and refines structures iteratively. **dyMEAN** [Kong et al., 2023] generates designs with full-atom modeling. **ABGNN** [Gao et al., 2023] introduces a pre-trained antibody language model combined with graph neural networks for one-shot sequence-structure generation. **DiffAb** [Luo et al., 2022] utilizes diffusion models to model type, position and orientation of each amino acid. **AbX** [Zhu et al., 2024] integrates evolutionary priors, physical interaction modeling, and geometric constraints into a score-based diffusion framework for sequence-structure co-design. All methods except for MEAN are capable of generating multiple antibodies for a specific antigen. To ensure a fair comparison, we implement a random version of MEAN by adding a small amount of random noise to the input structure.

### 5.2 Antigen-binding CDR-H3 Design

**Evaluation Metrics.** To better measure the gap between designs generated by different models and natural antibodies, we use CDR $E_{\text{total}}$ and CDR-Ag $\Delta G$ as defined above, rather than commonly used metrics such as AAR and RMSD. Additionally, we include CDR-Ag $E_{\text{att}}$ and CDR-Ag $E_{\text{rep}}$ used during model alignment. Zhou et al. [2024b] argue that these physics-based measurements are indispensable in designing nature-like antibodies and act as better indicators of the rationality and functionality of antibodies. Based on energy measurements, we compute energy gap as the mean absolute error relative to natural antibodies. We sample 1,280 antibodies using each method and perform structure refinement with the relax protocol in Rosetta [Chaudhury et al., 2010]. To select the best sample from each test case, we aggregate rankings of CDR $E_{\text{total}}$ and CDR-Ag $\Delta G$.

**Results.** We report the main results in Table 2. We also include metrics for RMSD and AAR in Table 4 and additional binding and developability metrics in Table 5. We present visualization examples in Figure 4. Overall, AlignAb outperforms baseline methods and narrows the gap between generated and natural antibodies. Furthermore, AlignAb demonstrates the smallest difference between top samples and average samples, suggesting a higher consistency in the generated antibody quality.

While baseline methods possess lower values for certain energy measurements, the generated antibodies are often far from ideal. For instance, MEAN, despite achieving a low CDR-Ag $\Delta G$, exhibits significantly higher CDR $E_{\text{total}}$, indicating less favorable overall interactions and potential structural clashes. HERN, dyMEAN and ABGNN show poor performance across most metrics, with high CDR $E_{\text{total}}$ values, suggesting strong repulsion due to close antigen-antibody proximities. Comparatively, DiffAb demonstrates a more balanced approach. It benefits from the theoretically guaranteed diversity of diffusion models and produces a higher variance in the quality of the designed CDRs. This provides DiffAb a higher probability of generating high-quality top-1 designs compared to other baselines. AbX further improves over DiffAb by incorporating evolutionary, physical, and geometric constraints, producing antibodies with more realistic structures and better binding energies. However, AlignAb surpasses all baselines, including AbX, showcasing smallest energy gap relative to natural antibodies. Through its energy alignment, AlignAb reduces average CDR $E_{\text{total}}$, CDR-Ag $\Delta G$ and CDR-Ag $E_{\text{rep}}$ by a large margin, while maintaining reasonable CDR-Ag $E_{\text{att}}$ values. This indicates antibodies generated by AlignAb have fewer clashes and exhibit strong binding affinity to target antigens.

Table 3: Ablation studies for the proposed AlignAb framework. Lower values are better in all measurements. Results show that each component contributes to improved antibody quality, with the full method (Iterative POEA) achieving the lowest energy gap and strongest overall performance.

| Method | CDR $E_{\text{total}} \downarrow$ | CDR–Ag $\Delta G \downarrow$ | CDR–Ag $E_{\text{att}} \downarrow$ | CDR–Ag $E_{\text{rep}} \downarrow$ | Gap $\downarrow$ |
|---|---|---|---|---|---|
| Reference | -11.03 | 16.75 | -12.45 | 15.77 | – |
| w/o Alignment | 128.10 | 235.82 | -14.31 | 479.00 | 205.82 |
| w/o Pre-training | 50.23 | 76.89 | -13.84 | 118.29 | 56.33 |
| DPO | 113.59 | 183.19 | **-19.98** | 352.15 | 158.74 |
| POEA | 48.53 | 74.85 | -13.41 | 103.54 | 51.59 |
| **Iterative POEA** | **34.13** | **55.71** | -14.60 | **59.06** | **32.39** |

We anticipate further performance improvements beyond current results with several simple modifications. Due to limited computational resources, we assign the same weight to the reward models across all test data (see Appendix E.2). By tuning the reward weightings, we are able to optimize the energy trade-offs between multiple conflicting objectives for each antigen-antibody complex, potentially resulting in an improved Pareto front of optimized models. Additionally, increasing the sample size and number of iterations for alignment will likely enhance the overall performance and reliability of the generated antibodies. These preliminary results underscore the potential of AlignAb in generating nature-like antibodies. We include the full evaluation results in Table 6.

## 5.3 Ablation Studies

Our approach combines a three-stage training pipeline (pre-training, supervised fine-tuning, and preference alignment), a Pareto-Optimal Energy Alignment objective, and an iterative online exploration strategy with temperature scaling. To quantify the contribution of each component, we evaluate variants of AlignAb on the first 10 antigens in the test set by generating 32 candidates for each antigen and report the average energy metrics. We present quantitative results in Table 3 and qualitative examples for a specific antigen with PDB ID 5nuz in Figure 2.

**Three-Stage Training.** We assess the contribution of each stage using the metrics described in Section 5.2. We observe that the variant without alignment stage performs poorly across all energy metrics as indicated by Table 3, highlighting that preference alignment is essential for steering the generator toward energetically favorable designs. The variant without pre-training stage, which disregards knowledge learned from large-scale antibody pre-training, also degrades performance relative to the full method. This confirms that pre-training provides a strong inductive bias that yields more realistic and stable initial proposals before alignment. Taken together, these results indicate that pre-training, supervised fine-tuning, and preference alignment are mutually reinforcing and are all essential components of a robust pipeline for high-quality antibody design.

**Pareto-Optimal Energy Alignment.** We evaluate the impact of the proposed POEA algorithm through controlled ablations. As shown in Table 3, the DPO baseline, which follows the original DPO formulation, performs significantly worse than our proposed method. This validates that our multi-objective POEA function is substantially more effective at balancing competing energy terms than a naive DPO approach. We also compare our full, iterative method to a single-iteration variant. The iterative method consistently achieves the highest performance across all metrics. This demonstrates that our iterative online exploration paradigm, where the model refines its policy over multiple turns, is critical for converging to the most optimal designs. This aligns with our analysis in Figure 2, which shows that the full framework with online exploration and temperature scaling produces a superior Pareto front compared to offline alignment or alignment without temperature scaling.

**Balancing Conflicting Objectives.** One of the central challenges in antibody design is resolving conflicting biophysical objectives, such as maximizing binding affinity (low $E_{\text{att}}$) and minimizing steric clashes (low $E_{\text{rep}}$). Figure 2 visualizes this trade-off by plotting the Pareto front of these two energy values. Our full method, AlignAb, consistently occupies the lower-left region of the plot, maintaining a smaller gap to the reference antibody (yellow) compared to baseline methods like DiffAb and other alignment variants. This position indicates that AlignAb achieves a superior balance and lower overall energy. The figure also illustrates the typical failure modes that arise from unbalanced objectives. When attractive energy $E_{\text{att}}$ is weighted too strongly in Figure 2 (C), the model overemphasizes bulky aromatic residues such as Tyrosine and Tryptophan, causing steric clashes and elevated $E_{\text{rep}}$. Conversely, over-weighting the repulsive term $E_{\text{rep}}$ in Figure 2 (D) biases the model toward small residues like Serine and Glycine, which results in weak binding affinities.

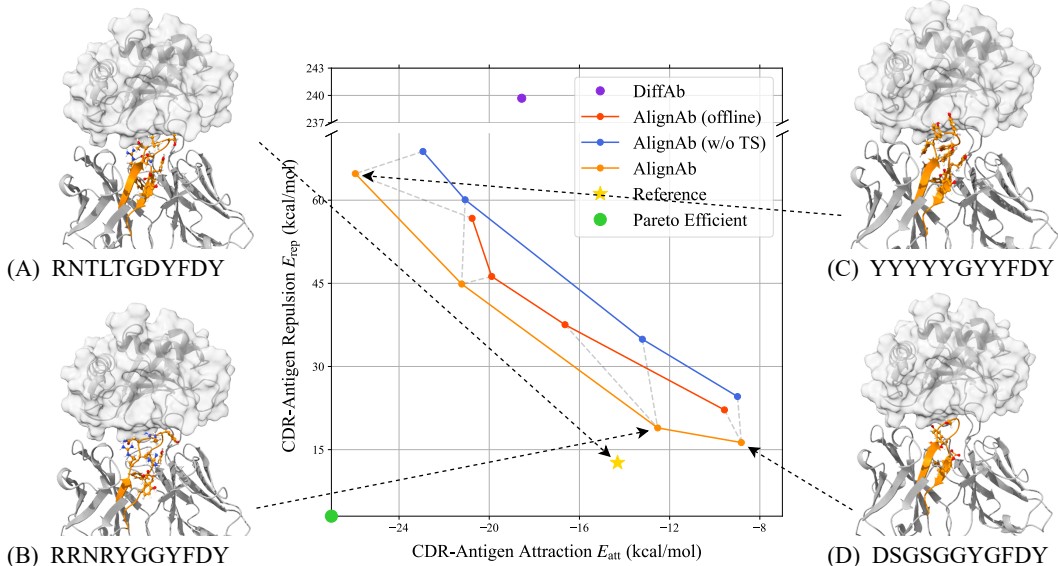

(A) RNTLTGDYFDY  (C) YYYYYGYYFDY

(B) RRNRYGGYFDY  (D) DSGSGGYGFDY

Figure 2: Frontiers of CDR-Ag $E_{att}$ and CDR-Ag $E_{rep}$ alignment and typical samples produced by different reward weightings in POEA. **(A)** is the reference CDR-H3 (colored in orange) from PDB ID 5nuz. **(B)** is the best CDR-H3 design generated by AlignAb with low overall energy and high similarity with the reference structure. **(C)** is the typical type of design when $E_{att}$ reward dominates, and often consists of large side chains and contains structural collisions. **(D)** is the typical type of design when $E_{rep}$ reward dominates, and often lack of side chains with weak binding with the antigen.

In contrast, our multi-objective POEA formulation is able to identify Pareto-optimal solutions in Figure 2 (B) that achieve both strong attraction and low repulsion. This allows our method to produce stable, high-affinity antibody designs that closely resemble the reference structure in Figure 2 (A), This demonstrates that AlignAb effectively balances conflicting objectives through Pareto-guided optimization, leading to nature-like antibodies with well-formed binding interfaces.

# 6   Conclusion

In this work, we adapt the successful paradigm of training Large Language Models to the task of antibody sequence-structure co-design. Our three-stage training pipeline addresses the key challenges posed by limited structural antibody-antigen data and the common oversight of energy considerations during optimization. During alignment, we optimize the model to favor antibodies with low repulsion and high attraction to the antigen binding site, enhancing the rationality and functionality of the designs. To mitigate conflicting energy preferences, we extend AbDPO in combination with iterative online exploration and temperature scaling to achieve Pareto optimality under multiple alignment objectives. Our proposed methods demonstrate high stability and efficiency, producing a superior Pareto front of antibody designs compared to top samples generated by baselines and previous alignment techniques. Future work includes further investigating the performance of the framework using larger fine-tuning datasets and extending our method to other structures such as small molecules.

## Acknowledgments

The authors would like to thank Abhishek Pandey and Weimin Wu for valuable conversations and Jiayi Wang for coordinating computational resources. The authors would like to thank the anonymous reviewers and program chairs for constructive comments.

Han Liu is partially supported by NIH R01LM1372201, NSF AST-2421845, Simons Foundation MPS-AI-00010513, AbbVie, Dolby and Chan Zuckerberg Biohub Chicago Spoke Award. This research was supported in part through the computational resources and staff contributions provided for the Quest high performance computing facility at Northwestern University which is jointly supported by the Office of the Provost, the Office for Research, and Northwestern University Information Technology. The content is solely the responsibility of the authors and does not necessarily represent the official views of the funding agencies.

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

# Appendix

# A  Supplementary Backgrounds

## A.1  Diffusion Processes for Antibody Generation

A diffusion probabilistic model consists of two processes: the forward diffusion process and the reverse generative process. Let $T$ denote the terminal time, and $t \in [T]$ denote the diffusion time step. Let $\mathcal{R}^t = \{(s_j^t, \boldsymbol{x}_j^t, \boldsymbol{O}_j^t) \mid j = l+1, \ldots, l+m\}$ denote a sequence of latent variables sampled during the diffusion process, where $(s_j^t, \boldsymbol{x}_j^t, \boldsymbol{O}_j^t)$ is the intermediate state for amino acid $j$ at diffusion step $t$. Intuitively, the forward diffusion process injects noises to the original data $\mathcal{R}^0$, while the reverse generative process learns to recover ground truth by removing noise from $\mathcal{R}^T$. To model both the sequence and structure of antibodies, Luo et al. [2022] defines three separate diffusion processes for $q(\mathcal{R}^t \mid \mathcal{R}^0)$ as follows:

$$q(s_j^t \mid s_j^0) = \mathcal{C}\Big(\mathbb{1}(s_j^t) \,\Big|\, \bar{\alpha}^t \cdot \mathbb{1}(s_j^0) + (1 - \bar{\alpha}^t) \cdot \frac{1}{20} \cdot \mathbf{1}\Big),$$

$$q(\boldsymbol{x}_j^t \mid \boldsymbol{x}_j^0) = \mathcal{N}\Big(\boldsymbol{x}_j^t \,\Big|\, \sqrt{\bar{\alpha}^0} \cdot \boldsymbol{x}_j^0, (1 - \bar{\alpha}^0)\boldsymbol{I}\Big),$$

$$q(\boldsymbol{O}_j^t \mid \boldsymbol{O}_j^0) = \mathcal{IG}_{\mathrm{SO}(3)}\Big(\boldsymbol{O}_j^t \,\Big|\, \mathsf{ScaleRot}\big(\sqrt{\bar{\alpha}^t}, \boldsymbol{O}_j^0\big), 1 - \bar{\alpha}^t\Big),$$

where $\bar{\alpha}^t = \prod_{\tau=1}^t (1 - \beta^\tau)$ and $\{\beta^t\}_{t=1}^T$ is the predetermined noise schedule. Here, $\mathcal{C}$ denotes the categorical distribution defined on 20 types of amino acids; $\mathcal{N}$ denotes the Gaussian distribution on $\mathbb{R}^3$; $\mathcal{IG}_{\mathrm{SO}(3)}$ denotes the isotropic Gaussian distribution on SO(3). We use $\mathbb{1}$ to represent one-hot encoding function and $\mathsf{ScaleRot}$ to represent rotation angle scaling under a fixed axis.

To recover $\mathcal{R}^0$ from $\mathcal{R}^T$ given specified antibody-antigen framework $\mathcal{F}$, Luo et al. [2022] defines the reverse generation process $p(\mathcal{R}^{t-1} \mid \mathcal{R}^t, \mathcal{F})$ at each time step as follows:

$$p(s_j^{t-1} \mid \mathcal{R}^t, \mathcal{F}) = \mathcal{C}\Big(s_j^{t-1} \,\Big|\, f_{\boldsymbol{\theta}_s}(\mathcal{R}^t, \mathcal{F})[j]\Big),$$

$$p(\boldsymbol{x}_j^{t-1} \mid \mathcal{R}^t, \mathcal{F}) = \mathcal{N}\Big(\boldsymbol{x}_j^{t-1} \,\Big|\, f_{\boldsymbol{\theta}_x}(\mathcal{R}^t, \mathcal{F})[j], \beta^t \boldsymbol{I}\Big),$$

$$p(\boldsymbol{O}_j^{t-1} \mid \mathcal{R}^t, \mathcal{F}) = \mathcal{IG}_{\mathrm{SO}(3)}\Big(\boldsymbol{O}_j^{t-1} \,\Big|\, f_{\boldsymbol{\theta}_O}(\mathcal{R}^t, \mathcal{F})[j], \beta^t\Big),$$

where all three $f_{\boldsymbol{\theta}}$ are parameterized by SE(3)-equivariant neural networks and $f(\cdot)[j]$ denotes the output for amino acid $j$. Therefore, the training objective consists of three parts:

$$\mathcal{L}_s^t = \mathbb{E}_{\mathcal{R}^t \sim p}\Big[\frac{1}{m} \sum_{j=l+1}^{l+m} \mathbb{D}_{\mathrm{KL}}\big(q(s_j^{t-1} \mid s_j^t, s_j^0) \,\|\, p(s_j^{t-1} \mid \mathcal{R}^t, \mathcal{F})\big)\Big], \tag{A.1}$$

$$\mathcal{L}_{\boldsymbol{x}}^t = \mathbb{E}_{\mathcal{R}^t \sim p}\Big[\frac{1}{m} \sum_{j=l+1}^{l+m} \big\| \boldsymbol{x}_j^0 - f_{\boldsymbol{\theta}_x}(\mathcal{R}^t, \mathcal{F})[j] \big\|^2\Big], \tag{A.2}$$

$$\mathcal{L}_{\boldsymbol{O}}^t = \mathbb{E}_{\mathcal{R}^t \sim p}\Big[\frac{1}{m} \sum_{j=l+1}^{l+m} \big\| (\boldsymbol{O}_j^0)^\mathsf{T} f_{\boldsymbol{\theta}_O}(\mathcal{R}^t, \mathcal{F})[j] - \boldsymbol{I} \big\|_F^2\Big]. \tag{A.3}$$

Finally, the overall loss function is $\mathcal{L} = \mathbb{E}_{t \sim \mathrm{Uniform}(1, \cdots, T)}[\mathcal{L}_s^t + \mathcal{L}_{\boldsymbol{x}}^t + \mathcal{L}_{\boldsymbol{O}}^t]$. After training the model, we can use the reverse generation process to design CDRs given the antibody-antigen framework.

## A.2  Optimal Policy of Equivalent Reward Functions

We cite the following definition and lemmas from DPO [Rafailov et al., 2024]:

**Definition A.1.** We say that two reward functions $r(x, y)$ and $r'(x, y)$ are equivalent iff $r(x, y) - r'(x, y) = f(x)$ for some function $f$.

**Lemma A.1.** Under the Plackett-Luce, and in particular the Bradley-Terry, preference framework, two reward functions from the same class induce the same preference distribution.

**Lemma A.2.** Two reward functions from the same equivalence class induce the same optimal policy under the constrained RL problem.

### A.3 DPO for Diffusion Model Alignment

Here we review DPO for diffusion model alignment [Wallace et al., 2023]. By alignment, we mean to align the diffusion models with users' preferences.

Let $\mathcal{D} \coloneqq \{(x, y_w, y_l)\}$ be a dataset consisting an input/prompt $x$ and a pair of output from a preference model $p_{\mathrm{ref}}$ with preference $y_w \succ y_l$. Our goal is to learn a diffusion model $p_\theta(y \mid x)$ aligning with such preference associated with $p_{\mathrm{ref}}$. Let $T$ denote the diffusion terminal time, and $t$ denote the diffusion time step. Let $y^{1:T}$ be the intermediate latent variables and $R(y, y^{0:T})$ be the commutative reward of the whole markov chain such that

$$r(x, y^0) \coloneqq \mathbb{E}_{p_\theta(y^{1:T} \mid x, y^0)}[R(y, y^{0:T})].$$

Aligning $p_\theta$ to $p_{\mathrm{ref}}$ needs

$$\max_{p_\theta} \left\{ \mathbb{E}_{x \sim \mathcal{D}\, y^{0:T} \sim p_\theta(y^{0:T} \mid x)}[r(x, y^0)] - \mathrm{D}_{\mathrm{KL}}\left[ p_\theta(y^{0:T} \mid x) \mid p_{\mathrm{ref}}(y^{0:T} \mid x) \right] \right\}.$$

Mirroring DPO (3.2), we arrive a ELBO-simplified DPO objective for diffusion model [Wallace et al., 2023, Appendix S.2]:

$$\mathcal{L}_{\mathrm{DPO-Diffusion}}(p_\theta, p_{\mathrm{ref}})$$

$$\leq - \mathbb{E}_{\substack{(x_0^w, x_0^l) \sim \mathcal{D}, \\ t \sim \mathcal{U}(0, T), \\ x_{t-1}^w, t \sim p_\theta(x_{t-1}^w \mid x_t^w), \\ x_{t-1}^l, t \sim p_\theta(x_{t-1}^l \mid x_t^l)}} \log \sigma \left( \beta T \log \frac{p_\theta(x_w^{t-1} \mid x_w^t)}{p_{\mathrm{ref}}(x_w^{t-1} \mid x_w^t)} - \beta T \log \frac{p_\theta(x_l^{t-1} \mid x_l^t)}{p_{\mathrm{ref}}(x_l^{t-1} \mid x_l^t)} \right),$$

where $\mathcal{U}$ denotes uniform distribution, $\beta$ is KL regularization temperature. We remark this objective has a simpler form for empirical usage, see [Wallace et al., 2023, Eqn. 14].

## B Additional Numerical Experiments

This section provides supplementary results to support the main findings. Appendix B.1 reports additional metrics (AAR and RMSD) comparing our method to baselines. Appendix B.2 reports additional binding and developability metrics to illustrate the effect of alignment stage. Appendix B.3 presents ablation examples illustrating CDR-antigen interaction trade-offs. Appendix B.4 includes detailed evaluation results across benchmarks.

### B.1 Additional Evaluation Metrics

Table 4: Summary of AAR and RMSD metrics by our method and other baselines. We follow the default sampling settings from all baselines and use ranked top-1 samples generated by our method. AlignAb* indicates the AlignAb framework without the alignment stage.

| Metrics | HERN | MEAN | dyMEAN | ABGNN | DiffAb | AlignAb* | **AlignAb** |
|---|---|---|---|---|---|---|---|
| AAR ↑ | 33.17 | 33.47 | **40.95** | 38.3 | 36.42 | 37.65 | 35.34 |
| RMSD ↓ | 9.86 | 1.82 | 7.24 | 2.02 | 2.48 | 2.25 | **1.51** |

## B.2 Additional Binding and Developability Metrics

Table 5: Summary of Boltz-2 IPTM for binding correlation and TAP scores for developability. We report averages over 16 candidates for each of the first 10 test antigens. AlignAb* indicates the AlignAb framework without the alignment stage. Higher Boltz-2 IPTM and lower deviation from reference TAP scores are desirable.

| Method | Boltz-2 IPTM ↑ | PSH | PPC | PNC | SFvCSP |
|--------|---------------|-----|-----|-----|--------|
| Reference | 0.728 | 118.554 | 0.096 | 0.522 | -0.471 |
| AlignAb* | $0.669 \pm 0.118$ | $119.720 \pm 7.397$ | $0.085 \pm 0.068$ | $0.402 \pm 0.280$ | $0.070 \pm 1.384$ |
| AlignAb | $\mathbf{0.698 \pm 0.104}$ | $120.510 \pm 6.069$ | $0.073 \pm 0.049$ | $0.363 \pm 0.065$ | $-0.116 \pm 1.072$ |

## B.3 Additional Ablation Examples

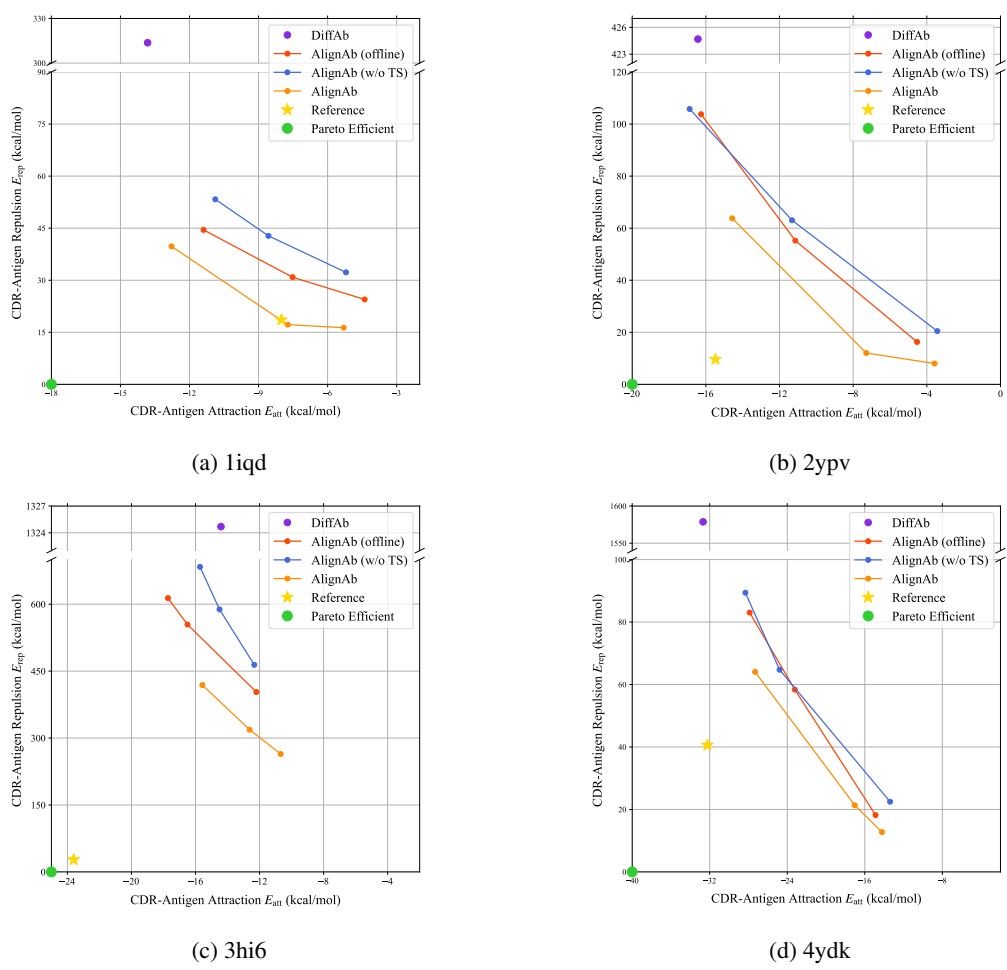

(a) 1iqd

(b) 2ypv

(c) 3hi6

(d) 4ydk

Figure 3: Frontiers of CDR-Ag $E_{\text{att}}$ and CDR-Ag $E_{\text{rep}}$ alignment produced by different reward weightings in POEA with four PDB examples.

## B.4 Detailed Evaluation Results

Table 6: Detailed evaluation results for various metrics for 42 antigens. The data source is the same as that in Table 2.

| PDB ID | DiffAb | | | | AlignAb | | | | MEAN | | | | ABGNN | | | | reference | | | |
|---|---|---|---|---|---|---|---|---|---|---|---|---|---|---|---|---|---|---|---|---|
| | Total | Nonrep | Rep | dG | Total | Nonrep | Rep | dG | Total | Nonrep | Rep | dG | Total | Nonrep | Rep | dG | Total | Nonrep | Rep | dG |
| 1a14 | -3.38 | -13.60 | 21.18 | 3.71 | 1.18 | -24.32 | 18.02 | -0.73 | 51.17 | -4.41 | 1.12 | -15.65 | 1,251.38 | -1.29 | 1.47 | -15.12 | -9.60 | -16.04 | 16.12 | -8.31 |
| 1a2y | -9.43 | -8.14 | 9.02 | -20.42 | -10.25 | -18.94 | 14.94 | -24.30 | 25.32 | -0.79 | 0.62 | -7.95 | 1,633.60 | -3.25 | 2.54 | -12.01 | -21.18 | -6.20 | 5.30 | -29.73 |
| 1fe8 | 2.15 | -5.78 | 9.33 | -6.52 | 0.37 | -14.42 | 9.66 | 20.40 | 1.74 | 0.00 | 0.44 | -9.59 | 1,054.35 | 0.00 | 0.00 | -9.92 | -18.24 | -18.11 | 15.38 | -21.40 |
| 1ic7 | 1.75 | -2.98 | 5.26 | 104.22 | 3.11 | -1.42 | 4.22 | 101.33 | 9.07 | -0.01 | 0.15 | -4.83 | 843.27 | 0.00 | 0.00 | -15.09 | -7.06 | -3.66 | 4.00 | 102.42 |
| 1iqd | 5.10 | -6.10 | 17.79 | -5.85 | 3.98 | -7.70 | 7.65 | -21.13 | 16.70 | -0.52 | 4.01 | -28.84 | 951.63 | -1.05 | 0.00 | -19.38 | -6.66 | -8.01 | 18.59 | 19.10 |
| 1n8z | 2.20 | -17.08 | 30.05 | 32.23 | 6.24 | -23.09 | 17.58 | 21.00 | 44.01 | -4.83 | 5.05 | -9.70 | 992.59 | 0.00 | 0.00 | 12.11 | 3.47 | -20.73 | 15.93 | 39.58 |
| 1ncb | 9.64 | -12.98 | 30.98 | 73.88 | 1.66 | -7.10 | 13.61 | 54.52 | 42.37 | -1.43 | 8.18 | -29.56 | 656.04 | -4.36 | 4.78 | -7.83 | -4.68 | -12.68 | 20.43 | 57.92 |
| 1osp | -0.97 | -11.40 | 18.56 | -10.47 | -7.06 | -17.39 | 26.50 | -8.22 | 69.34 | -9.59 | 11.47 | -80.86 | 1,044.68 | -2.54 | 2.51 | -6.51 | -23.03 | -14.41 | 19.38 | -19.06 |
| 1uj3 | -4.79 | -9.68 | 23.43 | 21.18 | -5.29 | -12.96 | 17.67 | 13.95 | -2.67 | 0.15 | 3.42 | -23.86 | 1,024.76 | -2.90 | 1.15 | 3.71 | -12.27 | -12.22 | 26.79 | 10.25 |
| 2adf | -5.15 | -8.72 | 25.37 | -15.88 | -1.59 | -10.66 | 20.15 | -25.67 | 14.58 | -1.92 | 8.87 | -14.37 | 1,216.99 | 0.00 | 0.00 | -20.54 | -26.27 | -25.54 | 16.20 | -41.24 |
| 2b2x | -3.00 | -12.04 | 43.50 | 2.58 | -2.62 | -23.63 | 27.30 | 3.18 | 23.52 | 0.84 | 6.67 | -10.04 | 1,216.00 | -0.91 | 9.73 | -13.26 | -16.57 | -12.61 | 20.90 | 9.67 |
| 2cmr | -9.02 | -21.28 | 21.21 | -11.00 | -10.12 | -19.82 | 22.29 | -13.98 | 32.28 | -8.56 | 4.04 | -30.25 | - | - | - | - | -28.71 | -15.28 | 15.31 | -16.29 |
| 2dd8 | -10.27 | -9.62 | 10.67 | 22.36 | -11.44 | -9.66 | 13.31 | 17.80 | 16.99 | -0.82 | 2.49 | -14.16 | 562.62 | 0.00 | 0.00 | -23.93 | -11.28 | -9.11 | 5.40 | 20.64 |
| 2vxt | 8.56 | -5.69 | 4.05 | -31.77 | 4.48 | -8.96 | 9.53 | -31.72 | 4.16 | 3.74 | 2.42 | -27.26 | 719.57 | -1.58 | 0.58 | 14.50 | -7.46 | -4.87 | 7.08 | -40.90 |
| 2xqy | -8.62 | -19.81 | 11.44 | -40.41 | -18.32 | -22.06 | 19.19 | -23.68 | 20.31 | -4.93 | 8.54 | -61.68 | 1,587.96 | -8.25 | 3.57 | -7.07 | -14.51 | -22.49 | 11.96 | -43.64 |
| 2xwt | 2.28 | -5.69 | 18.77 | -54.84 | -7.68 | -9.46 | 11.64 | -29.53 | 35.98 | -1.65 | 5.21 | -24.75 | 1,025.24 | -4.23 | 4.75 | -9.79 | -20.15 | -18.06 | 24.70 | -55.21 |
| 2ypv | 5.07 | -17.99 | 18.35 | -12.20 | 1.74 | -18.77 | 10.59 | -20.24 | 71.25 | -7.09 | 9.15 | 1.46 | 1,182.78 | -10.14 | 6.91 | -15.87 | -20.15 | -15.48 | 9.67 | -26.39 |
| 3hi6 | 1.04 | -2.12 | 7.37 | 34.10 | -6.46 | -15.12 | 10.50 | 12.76 | 49.90 | -8.21 | 8.38 | -5.44 | 1,235.52 | -2.38 | 0.56 | -11.22 | -18.35 | -23.61 | 27.54 | 8.76 |
| 3k2u | 4.81 | -5.71 | 32.82 | 39.42 | -1.55 | -6.28 | 17.17 | 33.81 | 2.73 | -0.82 | 0.58 | -31.30 | 1,024.09 | 0.00 | 0.00 | -21.06 | 3.70 | -23.11 | 49.67 | 19.41 |
| 3mxw | -2.61 | -6.74 | 5.77 | -6.01 | -7.36 | -8.00 | 11.15 | -10.50 | 32.38 | -6.52 | 7.30 | -6.42 | 1,597.12 | 0.74 | 2.98 | -10.47 | -13.94 | -6.21 | 14.25 | -20.09 |
| 3s35 | -7.41 | -2.85 | 10.14 | 15.79 | -4.80 | -1.38 | 2.20 | 12.01 | 11.89 | -0.91 | 4.62 | -25.42 | 1,040.04 | 0.00 | 0.00 | -19.82 | -17.63 | -5.99 | 5.68 | 35.18 |
| 4dvr | -11.21 | -15.90 | 4.42 | -13.20 | -8.25 | -17.83 | 6.92 | -20.41 | 51.42 | -5.04 | 3.63 | 10.37 | 1,176.08 | 0.00 | 0.00 | -21.90 | -25.13 | -12.64 | 6.76 | -11.78 |
| 4g6j | -5.27 | -9.85 | 26.69 | -18.91 | -9.29 | -7.98 | 8.44 | -18.32 | 14.18 | -0.29 | 7.58 | -4.91 | 899.87 | 0.00 | 0.00 | -1.05 | -10.34 | -11.76 | 17.83 | -29.98 |
| 4g6m | 0.76 | -20.55 | 17.45 | -22.17 | 1.02 | -17.70 | 17.99 | -22.13 | 30.97 | -4.17 | 6.67 | -15.02 | 1,788.05 | 0.00 | 0.00 | -9.14 | -14.16 | -25.05 | 19.82 | -42.91 |
| 4h8w | 0.57 | -11.11 | 17.42 | -19.39 | -3.59 | -15.08 | 13.11 | -14.90 | 27.28 | 1.51 | 7.53 | -19.58 | 1,737.00 | 0.46 | 0.58 | -14.34 | -17.13 | -13.80 | 26.20 | -21.68 |
| 4ki5 | -2.59 | -7.89 | 25.40 | -34.05 | -7.42 | -20.19 | 22.26 | -32.08 | 115.73 | -14.27 | 25.20 | -101.61 | 1,101.30 | 0.00 | 0.00 | -17.16 | -58.90 | -32.24 | 40.59 | -90.95 |
| 4lvn | 5.51 | -7.97 | 9.40 | 23.62 | 3.91 | -10.40 | 11.47 | 17.08 | 52.74 | -2.90 | 4.82 | 1.33 | 1,573.42 | 0.00 | 0.00 | -16.69 | -16.80 | -24.52 | 20.92 | 6.12 |
| 4ot1 | 6.86 | -17.43 | 24.19 | -35.54 | -18.19 | -26.24 | 18.46 | -70.11 | 168.30 | -6.16 | 5.82 | 4.23 | 1,937.80 | 0.06 | 2.23 | -9.41 | -59.10 | -27.58 | 19.46 | -64.09 |
| 4qci | -9.82 | -12.69 | 8.85 | -20.54 | -14.19 | -10.84 | 11.68 | -24.92 | 32.22 | -1.41 | 4.10 | 5.41 | 1,238.45 | -0.10 | 0.00 | -4.93 | -17.78 | -20.36 | 14.30 | -29.67 |
| 4xnq | -8.07 | -9.36 | 16.43 | -19.98 | -14.48 | -25.19 | 20.69 | -42.07 | 82.24 | -8.46 | 11.12 | -4.47 | 1,657.84 | 0.00 | 0.00 | -9.07 | -26.21 | -41.62 | 27.27 | -41.29 |
| 4ydk | -8.56 | -34.89 | 48.45 | 49.58 | -19.03 | -22.18 | 32.60 | -91.78 | 164.79 | -13.06 | 20.89 | -52.66 | 2,078.24 | 0.00 | 0.00 | -21.11 | -58.90 | -32.24 | 40.59 | -90.95 |
| 5b8c | 1.79 | -13.54 | 16.50 | -5.04 | -3.00 | -4.77 | 19.13 | 12.11 | 43.88 | -3.14 | 4.03 | -32.62 | 1,973.32 | -0.07 | 0.00 | -14.98 | -16.16 | -22.64 | 25.06 | -10.61 |
| 5bv7 | -9.80 | -28.35 | 19.92 | -86.88 | -14.73 | -21.44 | 21.01 | -30.26 | 107.85 | -21.52 | 28.63 | -17.58 | 1,885.45 | -2.79 | 0.84 | -5.94 | -32.05 | -16.66 | 10.44 | -77.33 |
| 5d93 | 2.43 | -11.97 | 5.37 | 54.56 | -6.70 | -12.74 | 8.06 | 46.18 | 9.57 | 0.00 | 4.64 | -17.40 | 931.27 | 0.00 | 0.00 | -14.07 | -5.68 | -13.48 | 11.39 | 41.54 |
| 5en2 | 0.88 | -18.47 | 19.38 | -40.37 | -8.18 | -20.55 | 17.19 | -42.05 | 107.25 | -18.51 | 16.29 | -16.98 | 1,470.04 | 1.46 | 0.00 | -15.06 | -42.87 | -25.64 | 11.91 | -52.88 |
| 5f9o | -3.31 | -14.59 | 10.80 | 19.61 | -10.45 | -20.54 | 16.57 | 27.75 | 102.81 | -17.36 | 28.27 | -22.90 | 1,449.84 | -3.62 | 3.39 | -25.94 | -15.83 | -17.66 | 16.28 | 17.82 |
| 5ggs | -2.20 | -14.62 | 36.29 | -16.02 | -10.87 | -15.57 | 16.80 | -21.65 | 22.19 | -10.58 | 8.92 | -12.95 | 1,813.13 | 0.00 | 0.00 | -3.88 | -26.34 | -21.97 | 18.72 | -32.84 |
| 5hi4 | 8.84 | -8.85 | 26.02 | -20.39 | 3.73 | -5.52 | 6.49 | 21.09 | 10.03 | 0.03 | 1.92 | -5.07 | 831.34 | -4.94 | 1.21 | -10.54 | -17.45 | -24.16 | 25.53 | -40.73 |
| 5j13 | -4.91 | -15.90 | 14.88 | -41.02 | -15.48 | -23.48 | 17.11 | 41.87 | 75.82 | -12.28 | 12.14 | -12.22 | 1,503.71 | 0.00 | 0.00 | -10.48 | -15.85 | -21.72 | 11.62 | -55.59 |
| 5l6y | -3.70 | -20.66 | 23.83 | -32.30 | -12.39 | -19.64 | 24.22 | -27.57 | 78.93 | 2.35 | 6.54 | -8.02 | 2,620.12 | 0.00 | 0.00 | -4.56 | -24.07 | -28.83 | 14.06 | -41.16 |
| 5mes | 0.73 | -9.85 | 12.47 | 11.27 | -7.95 | -10.62 | 9.69 | 1.96 | 26.99 | -3.57 | 5.08 | -19.04 | 1,167.09 | -5.76 | 0.64 | -12.15 | -20.00 | -17.20 | 10.25 | 1.53 |
| 5nuz | 0.18 | -6.00 | 23.26 | -27.38 | -20.09 | -15.74 | 27.22 | -35.10 | 45.09 | -18.48 | 9.93 | -35.39 | 1,235.44 | -9.42 | 857.92 | -15.24 | -31.07 | -14.31 | 12.65 | -54.31 |

# C  Additional Visualization

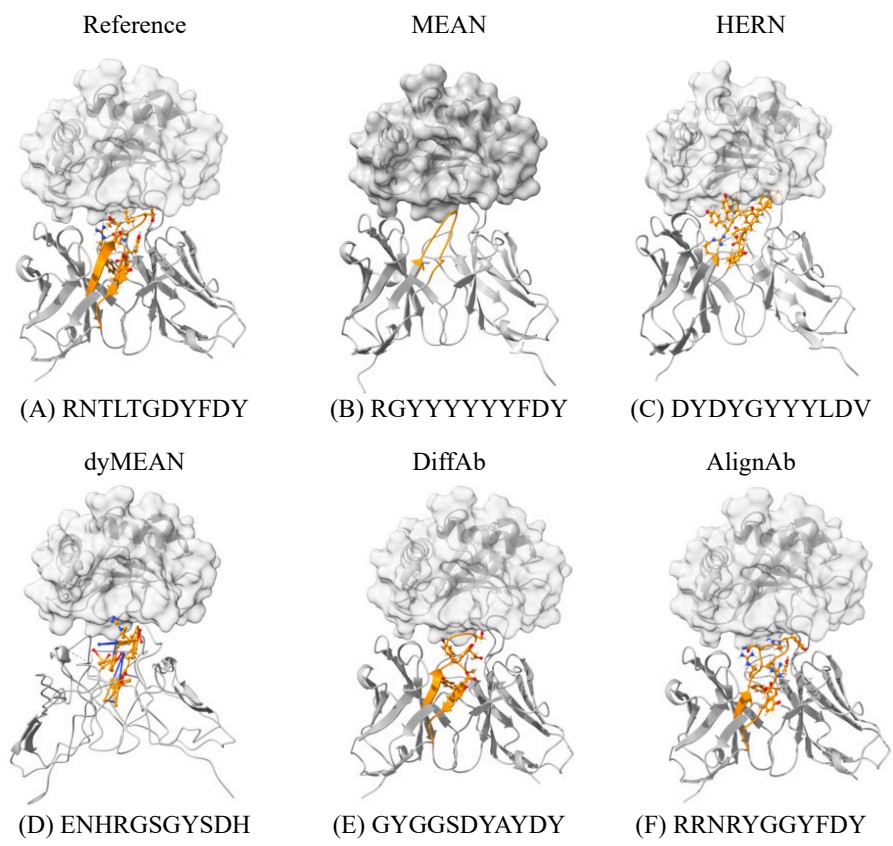

Figure 4: Visualization of reference antibody for antigen (PDB ID 5nuz) and different antibodies designed by our method and other baselines. The designed CDR-H3 structures are colored in orange and the designed CDR-H3 sequences are recorded accordingly.

## D    Energy Calculation and Reward Models

In Section 5, we introduce the calculation of two functionality-associated energies, CDR-Ag $E_{\text{att}}$ and CDR-Ag $E_{\text{rep}}$. Following Zhou et al. [2024b], we denote the residue with the index $i$ in the antibody-antigen complex as $A_i$. We then represent the **side chain** of the residue as $A_i^{sc}$ and **backbone** of the residue as $A_i^{bb}$, respectively.

We define the interaction energies between a pair of amino acids as EP, with the default weights defined by REF15 [Adolf-Bryfogle et al., 2017]. EP consists of six different energy types: $EP_{\text{hbond}}$, $EP_{\text{att}}$, $EP_{\text{rep}}$, $EP_{\text{sol}}$, $EP_{\text{elec}}$, and $EP_{\text{lk}}$. Following the settings from Section 3, we define the indices of residues within the CDR-H3 range from $l + 1$ to $l + m$, and the indices of residues within the antigen range from $g + 1$ to $g + n$. Thus, for the CDR residue with the index $j$, the CDR-Ag $E_{\text{att}}$ and CDR-Ag $E_{\text{rep}}$ are defined as:

$$\text{CDR-Ag } E_{\text{att}}^j = \sum_{i=g+1}^{g+n} \sum_{e \in \{\text{hbond,att,sol,elec,lk}\}} \Big( EP_{\text{e}}(A_j^{sc}, A_i^{sc}) + EP_{\text{e}}(A_j^{sc}, A_i^{bb}) \Big), \tag{D.1}$$

$$\text{CDR-Ag } E_{\text{rep}}^j = \sum_{i=g+1}^{g+n} \Big( EP_{\text{rep}}(A_j^{sc}, A_i^{sc}) + EP_{\text{rep}}(A_j^{sc}, A_i^{bb}) + 2 \times EP_{\text{rep}}(A_j^{bb}, A_i^{sc}) + 2 \times EP_{\text{rep}}(A_j^{bb}, A_i^{bb}) \Big). \tag{D.2}$$

From Equations (D.1) and (D.2), we conclude that the interaction energy between the CDR and the antigen is determined by both side-chain and backbone interactions. The CDR-Ag $E_{\text{att}}$ considers interactions primarily from side-chain atoms in the CDR-H3 region. In contrast, CDR-Ag $E_{\text{rep}}$ assigns higher costs to repulsions from backbone atoms in the CDR-H3 region. This reason for the different is that side-chain atoms contribute most of the interaction energy between CDR-H3 and the antigen, as shown in Figure 1. Therefore, CDR-Ag $E_{\text{att}}$ exhibits a benefit in interactions, while CDR-Ag $E_{\text{rep}}$ represents repulsive costs.

To guide the model alignment process, we utilize the above two energy definitions to compute the final rewards as follows:

$$r_{\text{att}}(x, y) = -\sum_{i=l+1}^{l+m} \text{CDR-Ag} E_{\text{att}}^j, \quad r_{\text{rep}}(x, y) = -\sum_{i=l+1}^{l+m} \text{CDR-Ag} E_{\text{rep}}^j, \tag{D.3}$$

where lower energy corresponds to a higher reward. Therefore, we compute the final collective reward with predetermined weights as $\hat{r}(x, y) = w_{\text{att}} r_{\text{att}}(x, y) + w_{\text{rep}} r_{\text{rep}}(x, y)$. We observe the repulsion reward is often several orders of magnitude bigger than the attraction reward. Therefore, we utilize the following reward margin in our actual experiments:

$$\Delta_{\hat{r}} = \log\big(\hat{r}(x, y_w) - \hat{r}(x, y_l)\big). \tag{D.4}$$

# E    Implementation Details

## E.1    Model Details

AlignAb consists of two parts: a pre-trained BERT model from AbGNN [Gao et al., 2023], and a pre-trained diffusion model from DiffAb [Luo et al., 2022]. For the pre-trained BERT model, our model uses a 12-layer Transformer model with a $\text{BERT}_{base}$ configuration. We set the embedding size to 768 and the number of heads to 12. For the pre-trained diffusion model, our model takes the perturbed CDR-H3 and its surrounding context as input. For example, 128 nearest residues of the antigen or the antibody framework around the residues of CDR-H3. The input consists of both single and pairwise residue embeddings. The number of features with single residue embedding is 128. It consists of the encoded information of its amino acid types, torsional angles, and 3D coordinates of all heavy atoms. The number of features with pairwise residue embedding is 64. It consists of the encoded information of the Euclidean distances and dihedral angles between the two residues. To combine the feature embeddings with the hidden representations learned from the pre-trained BERT model, we extract the embedding for each residue from the final layer of the BERT model and concatenate it with the single and pairwise residue embeddings. We then utilize multi-layer perception (MLP) neural networks to process the concatenated embeddings. The MLP has 6 layers. In each layer, the hidden state 128. The output of this neural network is the predicted categorical distribution of amino acid types, $C_\alpha$ coordinates, a $so(3)$ vector for the rotation matrix.

The diffusion models aim to generate amino acid types, $C_\alpha$ coordinates, and orientations. Hence, for training the diffusion models, we take the output of MLP as input for diffusion models. We set the number of diffusion (forward) stets to be 100. For the noise schedules, we apply the same setting of DDPM [Ho et al., 2020], utilizing a $\beta$ schedule with $s = 0.01$. The noises are gradually added to amino acid types, $C_\alpha$ coordinates, and orientations.

## E.2    Training Details

**Transferring.**    We train the diffusion model part of AlignAb following the same procedure as Luo et al. [2022]. The optimization goal is to minimize the rotation, position, and sequence loss. We apply the same weight to each loss during training. We utilize the Adam [Kingma and Ba, 2014] optimizer with `init_learning_rate=1e-4`, `betas=(0.9,0.999)`, `batch_size=16`, and `clip_gradient_norm=100`. We also utilize a learning rate scheduler, with `factor=0.8`, `min_lr=5e-6`, and `patience=10`. We perform evaluation for every 1000 training steps and train the model on one NVIDIA GeForce GTX A100 GPU, and it can converge within 36 hours and 200k steps.

**Alignment.**    After obtaining the diffusion model, we further align it with energy-based preferences provided by domain experts.    We utilize the Adam [Kingma and Ba, 2014] optimizer with `init_learning_rate=2e-7`, `betas=(0.9,0.999)`, `batch_size=8`, `clip_gradient_norm=100`. We set the KL regularization term $\beta = 100.0$. In each batch, we select 8 pairs of energy-based preference data with labeled rewards. We do not use learning rate scheduling during alignment stage. For rewards, we set the $w_{\text{att}}$ and $w_{\text{rep}}$ with a fixed ratio 1:3. In each alignment iteration, we fine-tune the diffusion model for 4k steps. We repeat this process 3 times for each antigen in the test set.

# F    Broader Impact

This research advances the intersection of machine learning and therapeutic antibody design by providing a generalizable framework for sequence-structure co-design. The proposed method not only enhances the rationality and efficiency of antibody generation but also offers a scalable approach applicable to broader biomolecular engineering tasks. By enabling more accurate and interpretable AI-driven design pipelines, this work can accelerate the discovery of life-saving therapeutics and benefit researchers and practitioners developing biologics for global health and social good.

