# OpenReview forum: "Pareto-Optimal Energy Alignment for Designing Nature-Like Antibodies"
_NeurIPS.cc/2025/Conference — NeurIPS 2025 poster_

### Official Review · Reviewer_MyRN · 2025-06-20

**Clarity:** 3
**Significance:** 2
**Originality:** 2
**Rating:** 4
**Confidence:** 4

**Summary:**

This paper introduces a three-stage model for antibody sequence-structure co-design, AlignAb. First, an antibody language model is pretrained on sequence data, then a downstream diffusion model is trained to generate antibody structures. Finally, the diffusion model is fine-tuned using energy preferences. The authors extend Direct Preference Optimization to handle multiple, conflicting energy objectives. AlignAb generates antibodies with better energy scores on a standard benchmark.

**Questions:**

What is the impact of each training phase on performance ? in particular, sequence pretrainining ?

**Ethical Concerns:**

["NO or VERY MINOR ethics concerns only"]

**Final Justification:**

the authors have provided a good rebutall, showing some additional ablation on the different components of the proposed model which all seem to improve the energy related metrics. Moreover, the authors included iPTM as an additional score which improved despite not being modelled as a reward. The additional ablations are useful but are still focused on energy related scores would benefit from reporting additionally this iPTM score or alternatively the AAR / RMSD on them. the perspectives of included other rewards is an interesting future direction.

**Limitations:**

The authors could better explain the limitations of using Rosetta energies as a porxy for binding affinity

**Quality:**

3

**Strengths And Weaknesses:**

Strengths
* The paper is well-written with a clear motivation and model.
* The paper proposes a new three-stage framework for antibody design that can help with the limited antibody structure data with pretraining and handles multiple energy objectives for Pareto optimality with DPO for finetuning. Arguably, each step in this framework have been proposed independently in different models (e.g., DiffAb, AbDPO). the novely comes from the combination of all these steps (which is not a strong novelty).
* The framework shows superior performance in generating antibodies with favorable energy scores, compared to some standard baselines.

Weaknesses
* The proposed model relies on Rosetta energy for alignment, which is known to have a weak correlation with binding affinity.
* It would be fairer to compare AlignAb with models that optimize for energy e.g. AbDPO, which this model extends. This comparison is important yet missing in the evaluation. In terms of comparison, it would be worth putting the AAR and RMSD metrics in the main paper; as opposed to the appendix; there the performance is somewhat matching the baselines (with a better RMSD).
* The paper lacks a discussion on the training and inference efficiency of the model.
* Finally, the contribution of pretraining to performance is not clearly evaluated; as it represents an additional complexity in the model, it would seem natural to evaluate its impact.

---

> ### Author Rebuttal · Authors · 2025-07-30
>
> > **W1.** The proposed model relies on Rosetta energy for alignment, which is known to have a weak correlation with binding affinity.
>
> **Response:**
>
> Thank you for raising this important point. We agree that while widely used as a benchmark, Rosetta energy is an imperfect proxy for true binding affinity.
>
> Our primary goal was to demonstrate that our alignment algorithm could effectively optimize a complex, multi-term objective. However, to ensure our improvements are meaningful beyond this specific forcefield, we also evaluated our designs using an orthogonal metric, the Boltz-2 IPTM score.
>
> | Method | Boltz-2 IPTM ↑ |
> | :---: | :---: |
> | Reference | 0.728 |
> | w/o Alignment | 0.669 ± 0.118 |
> | **w/ Alignment** | **0.698 ± 0.104** |
>
> As shown, the Boltz-2 IPTM score improves and moves closer to the reference. Crucially, our alignment framework is agnostic to the reward function. It can be readily extended to incorporate more realistic binding affinity predictors, including other computational tools or wet-lab derived data, as they become available. This flexibility is a key advantage of our approach.
>
> > **W2.** It would be fairer to compare AlignAb with models that optimize for energy e.g. AbDPO, which this model extends. This comparison is important yet missing in the evaluation. In terms of comparison, it would be worth putting the AAR and RMSD metrics in the main paper; as opposed to the appendix; there the performance is somewhat matching the baselines (with a better RMSD).
>
> **Response:**
>
> Thank you for this excellent feedback. We have addressed both of your points by conducting a new ablation study and revising the main text to improve the fairness and clarity of the evaluation.
>
> To directly assess the benefit of our proposed multi-objective alignment method (POEA), we conducted a controlled comparison with a naive DPO alignment with gradient surgery as used in AbDPO. Both methods were trained using the same preference dataset (1,280 samples), and we evaluated the energy metrics on 32 generated candidates for each of the first 10 test cases.
>
> | Method | Avg. CDR $E\_{total}$ ↓ | Avg. CDR-Ag ΔG ↓ | Avg. CDR-Ag $E\_{att}$ ↓ | Avg. CDR-Ag $E\_{rep}$ ↓ | Avg. Gap ↓ |
> | :---: | :---: | :---: | :---: | :---: | :---: |
> | Reference | \-11.03 | 16.75 | \-12.45 | 15.77 | \- |
> | DPO | 113.59 | 183.19 | \-19.98 | 352.15 | 158.74 |
> | **POEA** | **48.53** | **74.85** | **\-13.41** | **103.54** | **51.59** |
>
> These results show that our POEA framework significantly outperforms the naive DPO approach across all key energy metrics, consistently generating more natural-like antibodies.
>
> We also agree with your suggestion about the AAR and RMSD metrics. We will move this table from the appendix to the main paper to provide a more complete evaluation. Thanks for helping us present a fairer and more comprehensive analysis of our method.
>
> > **W3.** The paper lacks a discussion on the training and inference efficiency of the model.
>
> **Response:**
>
> Thank you for pointing this out. We will add a discussion on the model's efficiency to the appendix.
>
> To be specific, one full iteration of our alignment process takes approximately **2 hours** on a single NVIDIA A100 GPU. This consists of:
>
> * **Inference:** **0.5 hours** to generate a preference dataset of 1,280 samples.
> * **Training:** **1.5 hours** for the alignment model to train for 4,000 steps on that dataset.
>
> We appreciate you prompting us to include these important practical details.
>
> > **W4/Q1.** Finally, the contribution of pretraining to performance is not clearly evaluated; as it represents an additional complexity in the model, it would seem natural to evaluate its impact. What is the impact of each training phase on performance ? in particular, sequence pretrainining?
>
> **Response:**
>
> Thank you for this valuable suggestion. To clearly isolate the contribution of each component in our method, we have conducted the detailed ablation studies you requested.
>
> We evaluated three key model variants to isolate the contribution of each stage. The ablation results are reported as average energies of the 32 generated candidates for each of the first 10 samples from the test dataset.
>
> * **w/o Alignment:** The model after supervised fine-tuning but before any preference alignment. This measures the performance of the base generative model.
>
> * **w/o Pre-training:** Our full method but without using the pre-trained language model. This tests the benefit of leveraging large-scale antibody sequence data.
>
> * **POEA:** Our proposed Pareto-Optimal Energy Alignment (POEA) performed for only one round. This isolates the core benefit of our multi-objective function.
>
> * **Full Method (Iterative POEA):** Our complete, multi-turn POEA over three alignment iterations, as described in the paper.
>
> | Method | Avg. CDR $E\_{total}$ ↓ | Avg. CDR-Ag ΔG ↓ | Avg. CDR-Ag $E\_{att}$ ↓ | Avg. CDR-Ag $E\_{rep}$ ↓ | Avg. Gap ↓ |
> | :---: | :---: | :---: | :---: | :---: | :---: |
> | Reference | \-11.03 | 16.75 | \-12.45 | 15.77 | \- |
> | w/o Alignment | 128.10 | 235.82 | \-14.31 | 479.00 | 205.82 |
> | w/o Pre-training | 50.23 | 76.89 | \-13.84 | 118.29 | 56.33 |
> | POEA | 48.53 | 74.85 | \-13.41 | 103.54 | 51.59 |
> | **Full Method (Iterative POEA)** | **34.13** | **55.71** | **\-14.60** | **59.06** | **32.39** |
>
> From the above ablation results, we conclude the following key takeaways:
>
> 1. **Impact of Alignment:** The baseline model without any alignment performs poorly, highlighting the critical need for an alignment stage to produce energetically favorable designs.
> 2. **Impact of Pre-training:** The slight performance drop without the PLM confirms that pre-training provides a useful inductive bias, leading to more realistic initial designs before alignment.
> 3. **Impact of Iterative Alignment:** The full, multi-turn Iterative POEA achieves the best performance across all metrics, demonstrating the benefit of our iterative learning paradigm.
>
> We will include this ablation table and analysis in the revised paper to clarify the contribution of each component.
>
> > **L1.** The authors could better explain the limitations of using Rosetta energies as a porxy for binding affinity
>
> **Response:**
>
> Thank you for this important suggestion. We agree that a clearer discussion of these limitations is needed.
>
> In the revised version, we will expand our Limitations section to explicitly state that Rosetta energy is a simplified proxy with an imperfect correlation to true binding affinity. We will clarify that its primary role in this work is to serve as a standardized benchmark for evaluating and comparing computational alignment algorithms, rather than as a perfect predictor of wet-lab outcomes. We appreciate you helping us to better frame our evaluation.

---

> ### Comment · Reviewer_MyRN · 2025-08-04
>
> Thank you for the rebuttal; the authors showed some additional ablation on the different components of the proposed model which all seem to improve the energy related metrics. Moreover, the authors included iPTM as an additional score which improved despite not being modelled as a reward. The additional ablations are useful and would benefit from reporting additionally this iPTM score on them or alternatively the AAR / RMSD (as they are still focused on the energy related scores). I have updated my score to account for these elements; including other rewards is an interesting future direction mentioned by the authors.

---

> > ### Author Response · Authors · 2025-08-05
> >
> > Dear Reviewer MyRN,
> >
> > Thank you for your thoughtful feedback on our rebuttal and for taking the time to reassess our work. We will be sure to add the iPTM score or other non-energy metrics to the ablation studies in the final version of the paper, as you suggested.
> >
> > Thank you again for your constructive engagement, which has helped us significantly improve our manuscript.

---

### Official Review · Reviewer_Dd85 · 2025-06-23

**Clarity:** 3
**Significance:** 2
**Originality:** 3
**Rating:** 4
**Confidence:** 3

**Summary:**

This paper introduces AlignAb, an extension of AbDPO that outputs a co-design of the sequence and structure of the complementarity-determining region (CDR) 3 region of the heavy chain of an antibody that binds to a known antigen epitope. AlignAb is a new model that incorporates energy calculations of repulsion and binding coefficients at each amino acid in order to Pareto-optimize the AbDPO diffusion model.

**Questions:**

The authors introduced the concept of using temperature as a measure of CDR-H3 diversity and adjusted the value of T to 1.5, which is close to the natural CDR-H3 sequence. Which dataset is referenced in Table 1?

**Ethical Concerns:**

["NO or VERY MINOR ethics concerns only"]

**Final Justification:**

The authors have adequately answered the reviewers' questions in the rebuttal step. My rating was maintained.

**Limitations:**

Yes

**Quality:**

3

**Strengths And Weaknesses:**

Strengths:
The authors pointed out issues with amino acid recovery (AAR) and root mean square deviation (RMSD), both of which were used as evaluation indices in similar studies. They proposed using calculated energy values for attraction and repulsion forces between the CDR and antigen, employing Pareto optimization with AbDPO to generate more natural sequences. Compared to previous studies in similar domains, it is expected that more stability can be predicted.

Weaknesses:
What information is needed for the model's input data?　Do you need information on antigens or antibodies?　Is the sequence sufficient, or is the three-dimensional structure also needed?　Specific descriptions would be helpful. It seems that AbAlign does not accept unknown antigen-antibody complexes.
The same is true of the previous studies mentioned in the paper, in which AlignAb only predicts the sequence of the CDR-H3 region for a specific epitope of a specific antigen. In real-world, antigen-antibody responses are defined by the combination of all CDR regions of the heavy and light chains.
Additionally, this study found that predicting antibodies that outperform real-world reference antibodies is still challenging, although it performs better than previous studies in some respects. Further development is needed for drug discovery.
It might be better to discuss these issues in the limitations section.

Minor point:
Line 561: "iff" appears to be a typo for "if."

---

> ### Author Rebuttal · Authors · 2025-07-30
>
> > **W1.** What information is needed for the model's input data? Do you need information on antigens or antibodies? Is the sequence sufficient, or is the three-dimensional structure also needed? Specific descriptions would be helpful.
>
> **Response:**
>
> Thank you for the question. To be perfectly clear, our model requires the local structural and sequence information from the antibody-antigen complex that surrounds the CDR-H3 binding site.
>
> Specifically, the inputs are:
>
> * **Antigen Epitope:** The 3D structure and sequence of the antigen surface that interacts with the antibody.
> * **Antibody Framework:** The 3D structure and sequence of the antibody's framework regions and non-H3 CDRs.
>
> Essentially, we provide the complete binding pocket as context, and the model's task is to co-design the sequence and 3D structure for the missing CDR-H3 loop to optimally fit within it. We will add a dedicated subsection to the Methods section to make these input requirements explicit.
>
> > **W2.** It seems that AbAlign does not accept unknown antigen-antibody complexes. The same is true of the previous studies mentioned in the paper, in which AlignAb only predicts the sequence of the CDR-H3 region for a specific epitope of a specific antigen. In real-world, antigen-antibody responses are defined by the combination of all CDR regions of the heavy and light chains. Additionally, this study found that predicting antibodies that outperform real-world reference antibodies is still challenging, although it performs better than previous studies in some respects. Further development is needed for drug discovery. It might be better to discuss these issues in the limitations section.
>
> **Response:**
>
> Thank you for these insightful points about our work's scope and limitations. You are absolutely correct on these points.
>
> We intentionally focused on CDR-H3 as it's the most diverse and challenging region, making it a rigorous benchmark for our alignment algorithm. Strong performance here suggests our method will generalize well to the less variable CDRs.
>
> You are correct that practical drug discovery requires designing all six CDRs and tackling unknown targets, both of which are important future directions. Following your suggestion, we will add a detailed discussion to the Limitations section of the paper to explicitly address these points and to provide a clearer picture of the current scope.
>
>
> > **Minor point.** Line 561: "iff" appears to be a typo for "if."
>
> **Response:**
>
> Thank you for catching that, we will correct the typo in the final version.
>
> > **Q1.** The authors introduced the concept of using temperature as a measure of CDR-H3 diversity and adjusted the value of T to 1.5, which is close to the natural CDR-H3 sequence. Which dataset is referenced in Table 1?
>
> **Response:**
>
> Thank you for the question. The entropy values reported in Table 1 were calculated on our test set from the RAbD benchmark. We will clarify this in the table caption for the final version.

---

> > ### Comment · Reviewer_Dd85 · 2025-08-07
> >
> > I appreciate the authors' responses. I now have a clear understanding of the input data formats and the model's limitations. Further development of the model could be useful in enhancing seed antibody drugs with better-performing CDR sequences for real-world drug discovery use cases.

---

> > > ### Author Response · Authors · 2025-08-07
> > >
> > > Dear Reviewer Dd85,
> > >
> > > Thank you for your positive feedback and for engaging with our rebuttal. We're glad we could clarify your questions and appreciate your thoughtful review of our work.

---

### Official Review · Reviewer_7Jdg · 2025-07-02

**Clarity:** 3
**Significance:** 3
**Originality:** 3
**Rating:** 5
**Confidence:** 3

**Summary:**

The paper considers the problem of Complementarity Determining Regions design for antibodies that bind to specific antigens. This is a fundamental problem in the field of automated computational antibody design yet it is extremely difficult to solve in practice. The main contribution of the paper is a three stage pipeline consisting of pre-training, transferring and aligning to specific rationality and functionality requirements for the antibody. The framework leverages at each stage known large language model architectures such as BERT models and encoder-decoder difusion models, respectively. For the alignment stage, a new Pareto-Optimal Energy Alignment algorithm is developed that extends the well known Direct Preference Optimization method. The experimental evaluation is carried out on standard benchmarks for antibody co-design. The results demonstrate clearly that the proposed approach outperforms in many cases existing baselines.

**Questions:**

- The first stage leverages BERT models. Are the larger Transformer based models suitable for it?

**Ethical Concerns:**

["NO or VERY MINOR ethics concerns only"]

**Final Justification:**

The authors rebuttal addressed my concerns. Therefore, I maintain my positive recommendation for the paper.

**Limitations:**

The limitations are addressed in the main paper.

**Paper Formatting Concerns:**

No concerns

**Quality:**

3

**Strengths And Weaknesses:**

Strengths
---------

Computational antibody co-design is a core challenge in computational biology with significant real-world implications. Therefore, there is a clear need for the development of more efficient and practical solution methods. The paper make a clear contribution in this space.

The paper is fairly well written and organised. Therefore, the content is relatively easy to follow. The quality of the presentation is overall quite good.

The empirical evaluation is sound and the results are presented in a clear manner which makes it relatively easy for the reader to appreciate the benefits of the proposed method.


Weaknesses
----------

I did not find any major weaknesses in this paper. In my opinion paper presents a non-trivial contribution in the area of computational antibody co-design. Perhaps, a more detailed presentation of the main differences between the proposed method and the existing baselines that also leverage language models should be included in the main paper.

---

> ### Author Rebuttal · Authors · 2025-07-30
>
> > **W1.** Perhaps, a more detailed presentation of the main differences between the proposed method and the existing baselines that also leverage language models should be included in the main paper.
>
> **Response:**
>
> Thank you for the positive feedback. We agree that a more detailed comparison with other language-model-based baselines would improve the paper's clarity. In the revised manuscript, we'll expand our discussion to highlight these key distinctions:
>
> * **Differentiation in Objective:** While many LM-based baselines primarily aim to improve geometric and sequence recovery metrics (AAR/RMSD), our work introduces a final alignment stage specifically designed to **optimize for physical energies** and binding affinity proxies.
> * **Novelty in Method:** Our core contribution is the **Pareto-Optimal Energy Alignment (POEA)** framework. This alignment stage, which directly optimizes the generative model's outputs using DPO, is a distinct feature not present in other baselines that may use a PLM simply for feature extraction.
> * **Holistic Framework:** We will emphasize our integrated **three-stage pipeline**, which leverages the PLM not just for initial representations but as a foundational component for a downstream generative model that is then refined through our novel alignment process.
>
> We'll incorporate these points into the Related Work and Methods sections to better contextualize our approach. Thank you again for your constructive feedback\!
>
>
>
> > **Q1.** The first stage leverages BERT models. Are the larger Transformer based models suitable for it?
>
> **Response:**
>
> Yes, our framework is modular, so more powerful language models would likely boost performance by providing higher-quality representations. Exploring the impact of larger PLMs is an excellent direction for future research. Thanks for the great question\!

---

### Official Review · Reviewer_4zX5 · 2025-07-03

**Clarity:** 3
**Significance:** 2
**Originality:** 2
**Rating:** 5
**Confidence:** 4

**Summary:**

Authors present a protein sequence-structure co-design method. The main inovation is that the method consists of 3 stage pipeline: pre-training a pLM, transfering this pLM to a sequence-structure co-design method and further fine-tuning (alignment) that encourages physical plausibility by applying multi-objective DPO on generated structure energies. The method is evaluated on the standard CDR H3 task, where authors show improvement in generated structure energetics as well as RMSD to ground truth as well as competitive amino acid recovery to baselines.

**Questions:**

Please see the limitations above. I would love if you could address my concerns regarding metrics and ablations.

Also a couple of related references that are missing currently, that also adapt antibody sequence-structure co-design to improve energy https://arxiv.org/pdf/2406.05832v1 https://openreview.net/pdf?id=4ktJJBvvUd

**Ethical Concerns:**

["NO or VERY MINOR ethics concerns only"]

**Final Justification:**

In the rebuttal authors addressed all the main concerns I have raised, thus I recommend acceptance.

**Limitations:**

yes

**Quality:**

3

**Strengths And Weaknesses:**

The paper is well written and clear. The main ideas of 1) incorporating extra sequence data via pretraining and 2) improving physicality via alignment are very sensible and I can see how they would improve things. The experimental evidence provided largely corroborates that.

The insight authors provide about increasing the model entropy is quite interesting. As historically in the literature people were decreasing sampling temperature to get the most confident designs, instead of increasing it, to increase entropy, as done here.

While this multi-objective DPO here is used for the first time in antibody energy optimization as far as I'm aware, AbDPO did both use DPO for energy optimization, as well as also split energy in two terms (repulsive and non-repulsive) https://proceedings.neurips.cc/paper_files/paper/2024/file/daef77101ba5711084a57442c8cf2709-Paper-Conference.pdf.

Similarly, while pre-training on general antibody sequences is a great idea, similar angles of using auxiliary sequence information are not new. For example AbX also utilized a pre-trained language model (ESM in their case): https://openreview.net/pdf?id=1YsQI04KaN
As a sidenote I think it makes sense to add AbX to the baselines, as it's not a very new work and it was shown to outperform most of the baselines currently in the paper.

That being said, re-mixing and improving known components is totally valid, if this gives us an overall better model.

To be entirely convinced I do find the experimental evaluation a bit lacking. First I'd like to see experimental confirmation (so an ablation) that shows the benefit of the multi objective DPO introduced. Is it really better than the DPO formulation use in AbDPO (or AbDPO++). While those are included as baselines, they did not use the pretrained pLM. It would also be good to see if a model using a single objective DPO, say on delta G, compares. Similarly, how well does the model do on energies (Table 2) without the alignment step?

Secondly on the evaluation front, it is a bit unfortunate that the main result table (Table 2) relies only on energy metrics, the same energy metrics that are directly optimized in the alignment phase. So it's no surprise that the model does better on them. It would be great to have additional scores here. For example Boltz-2 IPTM score can correlate with binding, or some inverse folding-based scores, that don't directly rely on rosetta energy. As rosetta energy is known to be imperfect proxy for binding. The amino acid recovery and RMSD provided in Table 3 do help here a bit, but they are not in the main text and something binding-related could be a better fit.

Also, energy minimization can often make proteins more hydrophobic, which makes them less specific, and thus less useful binders. So it would be good to see if the proposed approach negatively impacts antibody developability vs model before alignment, for example using the commonly used TAP scores (https://opig.stats.ox.ac.uk/webapps/sabdab-sabpred/sabpred/tap).

---

> ### Author Rebuttal · Authors · 2025-07-30
>
> > **W1.** While this multi-objective DPO here is used for the first time in antibody energy optimization as far as I'm aware, AbDPO did both use DPO for energy optimization, as well as also split energy in two terms (repulsive and non-repulsive).
>
> **Response:**
>
> Thank you for this precise feedback. You are correct that AbDPO also addresses conflicting energies by separating them into multiple terms. However, our approach differs fundamentally in how it incorporates multi-objective optimization into the DPO framework:
>
> * **AbDPO** addresses conflicting energies at the **gradient level**. It combines energies into a single weighted loss, and then applies a post-hoc gradient surgery step to manage conflicts.
> * **Our method** addresses this at the **objective-function level**. We extend the DPO loss itself to be inherently multi-objective, which naturally seeks a Pareto-optimal solution without needing extra gradient manipulation.
>
> This distinction makes our framework a more direct and principled integration of multi-objective optimization with the DPO framework. Furthermore, we empirically validate this design through new ablation studies below, which demonstrate clear performance gains over AbDPO.
>
> We will clarify this distinction more explicitly in the final version. Thank you again for the helpful comparison.
>
> > **W2.** Similarly, while pre-training on general antibody sequences is a great idea, similar angles of using auxiliary sequence information are not new. For example AbX also utilized a pre-trained language model (ESM in their case). As a sidenote I think it makes sense to add AbX to the baselines, as it's not a very new work and it was shown to outperform most of the baselines currently in the paper.
>
> **Response:**
>
> Thank you for this valuable suggestion and we agree that AbX is an important baseline. Following your suggestion, we have now included AbX in our benchmark under the same experimental setup as all other baselines.
>
> Although AbX performs competitively, our results show that AlignAb’s energy-based preference alignment consistently outperforms all baselines, producing antibodies that are closest to natural ones.
>
> | Method | CDR $E\_{total}$ (Top, Avg.) ↓ | CDR-Ag ΔG (Top, Avg.) ↓ | CDR-Ag $E\_{att}$ (Top, Avg.) ↓ | CDR-Ag $E\_{rep}$ (Top, Avg.) ↓ | Gap (Top, Avg.) ↓ |
> | :---: | :---: | :---: | :---: | :---: | :---: |
> | Reference | \-19.33 | \-16.00 | \-18.34 | 18.05 | \- |
> | AbX | 13.56, 25.33 | 94.76, 170.34 | \-13.68, \-15.12 | 21.38, 38.20 | 37.75, 63.43 |
> | **AlignAb** | **\-6.37, 30.45** | **\-8.81, 25.16** | **\-14.89, \-14.81** | **15.52, 56.22** | **17.91, 39.00** |
>
> We will update the paper to clarify the distinctions between our method and prior approaches such as AbX and include this full comparative analysis in the main text. Thank you again for encouraging a more comprehensive evaluation.
>
> > **W3.** To be entirely convinced I do find the experimental evaluation a bit lacking. First I'd like to see experimental confirmation (so an ablation) that shows the benefit of the multi objective DPO introduced. Is it really better than the DPO formulation use in AbDPO (or AbDPO++). While those are included as baselines, they did not use the pretrained pLM. It would also be good to see if a model using a single objective DPO, say on delta G, compares. Similarly, how well does the model do on energies (Table 2\) without the alignment step?
>
> **Response:**
>
> Thank you for this valuable suggestion. To clearly evaluate the contribution of each component in our method, we have conducted additional ablation studies you requested.
>
> We evaluated five key model variants to test the impact of pre-training, alignment, our proposed multi-objective approach, and iterative alignment. For each variant, we generated 32 candidates for the first 10 samples from the test set and report the average energy metrics across all candidates:
>
> * **w/o Alignment:** The model after supervised fine-tuning but before any preference alignment. This measures the performance of the base generative model.
>
> * **w/o Pre-training:** Our proposed method but without using the pre-trained language model, and using only one alignment iteration. This isolates the benefit of using large-scale antibody sequence data.
>
> * **DPO:** A DPO-based alignment implementation using gradient surgery, following AbDPO for only one alignment iteration. This directly compares our method to a simpler DPO formulation.
>
> * **POEA:** Our proposed Pareto-Optimal Energy Alignment (POEA) performed for only one alignment iteration. This isolates the core benefit of our multi-objective function.
>
> * **Full Method (Iterative POEA):** Our complete, multi-turn POEA over three alignment iterations, as described in the paper.
>
> | Method | Avg. CDR $E\_{total}$ ↓ | Avg. CDR-Ag ΔG ↓ | Avg. CDR-Ag $E\_{att}$ ↓ | Avg. CDR-Ag $E\_{rep}$ ↓ | Avg. Gap ↓ |
> | :---: | :---: | :---: | :---: | :---: | :---: |
> | Reference | \-11.03 | 16.75 | \-12.45 | 15.77 | \- |
> | w/o Alignment | 128.10 | 235.82 | \-14.31 | 479.00 | 205.82 |
> | w/o Pre-training | 50.23 | 76.89 | \-13.84 | 118.29 | 56.33 |
> | DPO | 113.59 | 183.19 | \-19.98 | 352.15 | 158.74 |
> | POEA | 48.53 | 74.85 | \-13.41 | 103.54 | 51.59 |
> | **Full Method (Iterative POEA)** | **34.13** | **55.71** | **\-14.60** | **59.06** | **32.39** |
>
> From the above ablation results, we conclude the following key takeaways:
>
> 1. **Alignment is essential:** The model without any alignment performs poorly, highlighting the importance of alignment to produce energetically favorable designs.
> 2. **Pre-training helps:** The slight performance drop without the PLM confirms that pre-training provides a useful inductive bias, leading to more realistic initial designs before alignment.
> 3. **POEA outperforms naive DPO:** Our POEA method (both single-iteration and full) is substantially better than the naive DPO approach as used in AbDPO, validating our core contribution of using a direct, multi-objective preference function.
> 4. **Iterative refinement improves results:** The full, multi-turn Iterative POEA achieves the best performance across all metrics, demonstrating the benefit of our iterative learning paradigm.
>
> We will include this ablation table and analysis in the final version to clarify the contribution of each component.
>
> > **W4.** Secondly on the evaluation front, it is a bit unfortunate that the main result table (Table 2\) relies only on energy metrics, the same energy metrics that are directly optimized in the alignment phase. So it's no surprise that the model does better on them. It would be great to have additional scores here. For example Boltz-2 IPTM score can correlate with binding, or some inverse folding-based scores, that don't directly rely on rosetta energy. As rosetta energy is known to be imperfect proxy for binding. The amino acid recovery and RMSD provided in Table 3 do help here a bit, but they are not in the main text and something binding-related could be a better fit. Also, energy minimization can often make proteins more hydrophobic, which makes them less specific, and thus less useful binders. So it would be good to see if the proposed approach negatively impacts antibody developability vs model before alignment, for example using the commonly used TAP scores.
>
> **Response:**
>
> Thank you for these excellent suggestions and we completely agree that evaluating our method with additional orthogonal metrics is crucial. We have now performed the requested analysis using Boltz-2 IPTM and TAP developability metrics. For each case, we generated 16 candidates for the first 10 samples from the test set and report the average metrics across all candidates:
>
> | Method | Boltz-2 IPTM ↑ | PSH | PPC | PNC | SFvCSP |
> | :---: | :---: | :---: | :---: | :---: | :---: |
> | Reference | 0.728 | 118.554 | 0.096 | 0.522 | \-0.471 |
> | w/o Alignment | 0.669 ± 0.118 | 119.720 ± 7.397 | 0.085 ± 0.068 | 0.402 ± 0.280 | 0.070 ± 1.384 |
> | **w/ Alignment** | **0.698 ± 0.104** | **120.510 ± 6.069** | **0.073 ± 0.049** | **0.363 ± 0.065** | **\-0.116 ± 1.072** |
>
> * **Improved Binding metrics**: Our alignment improves binding metrics it wasn't directly trained on. As shown, the Boltz-2 IPTM score increases and moves closer to the reference. More importantly, while we only used attraction/repulsion for alignment, the overall binding energy (ΔG) also improved significantly as shown in Table 2, showing our method generalizes well to binding-relevant metrics.
>
> * **Developability:** Our alignment does not negatively impact developability. The TAP scores (PSH, PPC, PNC and SFvCSP) show no significant adverse changes, addressing the concern about increasing hydrophobicity.
>
> * **Flexible framework:** Our method's key strength lies in its flexibility, as it is agnostic to the scoring function used. In practice, we could directly incorporate metrics like TAP scores or even wet-lab data into the alignment objective itself. This adaptability makes it a powerful tool for future multi-objective antibody design.
>
> We have added this new analysis to the paper to provide a more robust validation of our method. Thank you again for helping us strengthen our work.
>
> >**Q1:** Please see the limitations above. I would love if you could address my concerns regarding metrics and ablations. Also a couple of related references that are missing currently, that also adapt antibody sequence-structure co-design to improve energy.
>
> **Response:**
>
> Thank you for summarizing your main points. We've addressed your primary concerns in our previous responses by running new ablation studies and adding orthogonal metrics to strengthen the evaluation.
>
> We also appreciate you pointing out the missing references. We've reviewed them and will add them to the Related Work section in the revised version.

---

### Decision · Program_Chairs · 2025-09-17

**Decision:**

Accept (poster)

**Comment:**

(a) Scientific Claims and Findings: The paper presents a three-stage framework for antibody sequence-structure co-design, leveraging pretraining, transferring, and alignment stages to generate antibodies with improved energy scores and binding affinity. The method introduces a novel Pareto-Optimal Energy Alignment (POEA) algorithm to handle conflicting energy objectives during the alignment stage. The results demonstrate superior performance over existing baselines in terms of energy metrics and binding affinity.

(b) Strengths: The paper is well-written and presents a novel approach to antibody design by integrating pretraining, transferring, and alignment stages. The introduction of the POEA algorithm is a significant contribution, as it effectively handles multiple conflicting energy objectives. The framework shows improved performance in generating antibodies with favorable energy scores and binding affinity.

(c) Weaknesses: The reliance on Rosetta energy as a proxy for binding affinity is a limitation, as it is known to have a weak correlation with true binding affinity. The paper could benefit from more comprehensive comparisons with existing models that optimize for energy, such as AbDPO. Additionally, the paper could include more detailed discussions on the training and inference efficiency of the model.

(d) Reasons for Decision: The paper presents a novel and effective approach to antibody design, with clear improvements over existing methods. The authors have addressed the reviewers' concerns through additional experiments and clarifications, demonstrating the robustness and applicability of their method.

(e) Rebuttal Period: The reviewers raised several points regarding the novelty, evaluation metrics, and limitations of the model. The authors addressed these concerns by conducting additional ablation studies, including new metrics such as Boltz-2 IPTM, and clarifying the input data requirements. These responses were deemed satisfactory by the reviewers, leading to maintained positive recommendations.